# SpikeNet: Sparse Spike-Driven Mask Vector Transformer for Energy-Efficient and Stable Spiking Point Cloud Processing

## Abstract

The unordered nature of point cloud data poses significant challenges to conventional architectures primarily designed for structured data. Spiking neural networks (SNN), by virtue of their inherent sparsity and dynamics, are particularly well-suited for processing point clouds to effectively extract meaningful features. We propose SpikeNet, a novel spiking neural network architecture for energy-efficient and robust point cloud analysis. We introduce spiking-driven sparse attention mechanism coined the Spiking Vector Mask Transformer (SVMT). By dynamically aligning the sparsity of point cloud data through binary spiking masks, SVMT eliminates the need for softmax and multiplication operations, significantly improving computational efficiency. We also propose a Dynamic Sparse Spiking Residual (DSSR) structure and integrate it with SVMT to form the Spiking Neural Network (SpikeNet) for point cloud classification and segmentation. SpikeNet overcomes the trade-off between accuracy and efficiency in previous SNN methods, achieving collaborative optimization of performance and energy-efficiency. Experiments on benchmark datasets show that SpikeNet achieves state-of-the-art performance in shape classification and segmentation tasks, comparable to artificial neural network (ANN) based methods. *Our source code is in supplementary material and will be made publicly available.*

## 1 Introduction

Point cloud processing is a core technology for 3D perception. However, it faces critical challenges in real-time systems such as autonomous driving and robotics Vizzo et al. (2023); Lu et al. (2024); Liu et al. (2021); Liang et al. (2019); Ha et al. (2024). Despite significant advances in deep learning-based point cloud processing, existing methods remain constrained by the inherent trade-offs between data sparsity, disorder, and computational energy-efficiency. Traditional network architectures rely on dense computations or structured data representations, making them ill-suited for deployment on resource-constrained mobile platforms. High energy consumption not only shortens device battery life but also leads to thermal issues that can compromise system reliability. Therefore, developing a point cloud analysis method that balances efficient feature representation with energy optimization is essential for enabling practical deployment of 3D intelligence.

spiking neural networks (SNN), given their excellent energy efficiency, have attracted significant attention for 2D vision tasks Zhou et al. (2023b;a); Liu et al. (2024); Zhou et al. (2024b); Shi et al. (2024). Their event-driven computational paradigm offers a natural energy-efficiency advantage for 3D point cloud analysis. Different from the continuous activation mechanism of traditional artificial neural network (ANN), SNN transmit information through discrete spike sequences and only trigger sparse firings when the accumulated input reaches the neuron threshold. This dynamic characteristic naturally aligns with the sparsity of 3D point clouds where large blank areas would correspond to the non-activated states in SNN. SNN can suppress the neuron activities in blank (or highly sparse) areas and only generate spike activities in regions where point cloud data is significant, automatically avoiding unnecessary computations.

However, existing SNN-based point cloud analysis methods face dual challenges. On one hand, to pursue low energy consumption, existing methods often oversimplify the SNN architecture. This

compromises network accuracy, making it challenging to meet the stringent demands of applications such as autonomous driving. On the other hand, the non-differentiability of spiking signals leads to difficult training, limiting the generalization ability of the model. To address these issues, we propose SpikeNet. We design a spiking-driven sparse attention mechanism, the Spiking Vector Mask Transformer (SVMT), which dynamically aligns with the sparsity of point cloud data through binary spiking masks, eliminating the need for softmax and multiplication operations. This significantly improves computational efficiency. Additionally, we propose a Dynamic Sparse Spiking Residual structure (DSSR) to stabilize the gradient flow during SNN training. This prevents the degradation of deep features, and enhances training stability while maintaining energy-efficiency. Experiments show that SpikeNet significantly improves the accuracy of point cloud processing tasks while maintaining high energy efficiency. Our contributions are summarized below:

- We propose a spike-driven sparse attention mechanism coined Spiking Vector Mask Transformer (SVMT) that uses sparse spike-based queries, keys, and values, aligning dynamically to the sparsity of point cloud data through a binary spike mask. This approach eliminates the need for softmax and multiplication operations, significantly improving computational efficiency.

- We propose a Dynamic Sparse Spiking Residual (DSSR) structure and integrate it with the SVMT to form an SNN-based point cloud classification and segmentation network (SpikeNet). This approach addresses the training instability of SNN while maintaining energy efficiency, achieving collaborative optimization of high performance and low energy consumption.

- We design a directly-trained SNN point cloud segmentation solution. Experiments show that SpikeNet outperforms existing SNN methods on multiple benchmark datasets for point cloud classification and segmentation, and even achieves performance at par with the computationally expensive ANN-based models.

## 2 RELATED WORK

**ANN-based Point Cloud Analysis.** Point cloud processing methods can be broadly categorized into two approaches: direct processing of raw point clouds and projection onto voxel grids or 2D images. While projection-based methodsMaturana & Scherer (2015); Choy et al. (2019); Wei et al. (2020) simplify 3D tasks by converting them into 2D image problems, they often result in information loss, compromising the representation of fine-grained details. To address this, direct processing methods have gained prominence. PointNetQi et al. (2017a) pioneered the use of shared MLPs for unordered point sets, and PointNet++Qi et al. (2017b) further advanced this by introducing hierarchical feature learning to capture local geometric structures. PointNet++ has since become a cornerstone for modern point cloud analysis.

Recent research has focused on exploring local point features through graph-based, convolution-based, and attention-based methods. For instance, 3DGCNXia et al. (2021) constructs point clouds into graph structures to capture local geometric information via graph convolution operations. KP-ConvThomas et al. (2019) addresses the irregularity of point clouds by defining convolution kernels through sampled kernel points. PCTGuo et al. (2021) introduces attention mechanisms to compute weights between points, emphasizing key local features. Building on these advancements, our proposed SVMT module leverages attention mechanisms and the energy efficiency of SNN to enhance local feature extraction.

**SNN-based Point Cloud Analysis.** As a new energy-efficient neural network model, Spiking Neural Network (SNN) are still in early stages of application with point clouds. Spike PointNetRen et al. (2024) stands out, applying SNN to point clouds via a "trained-less but learning-more" paradigm, achieving strong multi-time-step inference with single-time-step training. P2SResLnetWu et al. (2024) combines spiking neurons with traditional point convolution, proposing a spatial-aware kernel point spiking neuron for 3D local perception. SPTWu et al. (2025) designs a queue-driven sampling direct coding for point clouds, building the first transformer-based point cloud classifier by integrating hybrid-driven neurons. E-3DSNNQiu et al. (2025) pioneered Spike Voxel Coding, encoding 3D point clouds into sparse spike sequences and extracting features efficiently via spike sparse convolution. However, these SNN models still lag significantly behind ANN in performance.

SNN show great promise in point cloud analysis due to event-driven computation and discrete spikes, minimizing redundant calculations and power use. Their core is spiking neurons (SNs), with Integrate-and-Fire (IF)Bulsara et al. (1996) and Leaky-Integrate-and-Fire (LIF)Gerstner & Kistler (2002) as the most common. IF neurons excel in energy-sensitive tasks for their low memory and energy needs. High-performance training is key for SNN development, with main methods being ANN-to-SNN conversion and direct training. The former converts ANN to spiking form but needs large time steps to simulate ReLU, causing notable delays. Inspired by ResNet, we found residual connections maintain SNN sparsity and performance, leading us to design a biologically plausible DSSR module to boost 3D point cloud processing. We also use direct training with IF neurons throughout, reducing latency and improving energy efficiency while ensuring performance.

## 3 METHOD

We propose SpikeNet, a point cloud analysis model based on Spiking Neural Network (SNN). SpikeNet integrates spiking-driven principles into residual architectures and point cloud transformers, to achieve energy-efficient feature learning. Below, we detail each component of SpikeNet and explain its architecture designed for 3D point cloud classification and segmentation.

### 3.1 DYNAMIC SPARSE SPIKING RESIDUAL

In current neural network research, enhancing computational efficiency and feature extraction capabilities stand as pivotal objectives. Drawing inspiration from the field of image classification Zhou et al. (2023b;a), we introduce a Dynamic Sparse Spiking Residual structure (DSSR) tailored for 3D point cloud analysis. As illustrated in Figure1, this module integrates two major advantages: a residual structure to address the vanishing gradient problem and the spiking neural network design to capture the temporal dynamic features. This enables us to precisely extract the features of 3D point clouds with a concise architecture. The core principle can be articulated as follows:

$$Y_1^{(l)} = \text{BN}(\text{Conv}(\text{SN}(X_1^{(l)}))) + X_1^{(l)}, \tag{1}$$

$$Y^{(l)} = \text{BN}(\text{Conv}(\text{SN}(Y_1^{(l)}))) + Y_1^{(l)}. \tag{2}$$

In our design, $X_1^{(l)}$ represents floating-point features, which transform into spike sequences after passing through the Spiking Neuron (SN) layer. The subsequent Convolution-Batch Normalization (Conv-BN) layer strictly confines its internal computations to floating-point additions, akin to the floating-point addition mechanism in spiking neural networks. This mechanism, being one of the fundamental operations in spiking neural networks, ensures computational stability and predictability. The residual structure then sums these outputs to yield $Y_1^{(l)}$, $Y_1^{(l)}$ as an output, remains in floating-point form and subsequently passes through another Spiking Neuron (SN) layer to convert the features into spike form. This conversion better captures the dynamic information of the data, aligning with the event-driven nature of spiking neural networks. Similarly, $Y^{(l)}$, also in floating-point form, serves as the output of the DSSR. Before integrating into the next module, it converts into a spike sequence, adhering to the computational rules of spiking neural networks.

### 3.2 SPIKING VECTOR MASK TRANSFORMER

Point Cloud TransformerGuo et al. (2021) has demonstrated the advantages of using offset attention mechanisms over the original self-attention mechanisms in point cloud processing. Meanwhile, Point TransformerZhao et al. (2021) has proven that vector attention operators are more effective than other operators in the field of 3D point cloud processing. Building on this, we have integrated the Spike mechanism and designed the Spiking Vector Mask Transformer (SVMT) as shown in Figure 2. The SVMT module consists of two parts: the Spiking Vector Attention (SVA) and the Spike Point Masker (SPM).

#### 3.2.1 SPIKING VECTOR ATTENTION

For the given input feature $S^{(l-1)}$, we first transform it into a spike sequence $S'^{(l-1)}$ with a value range of $\{0,1\}$ through a Spiking Neural Network (SNN) layer. Subsequently, the query matrix $Q$, key matrix $K$, and value matrix $V$ are computed via linear transformations. These matrices are

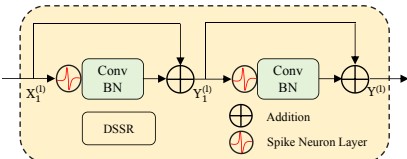

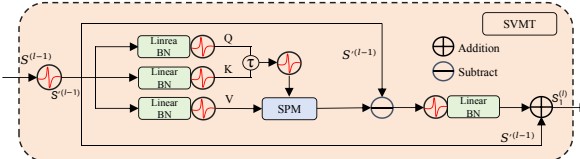

Figure 1: The design of the Dynamic Sparse Spiking Residual structure.

Figure 2: The design of the Spiking Vector Mask Transformer.

then processed by different spiking neurons. We denote the sequential execution of Linear, batch normalization, and spiking layers as LBS and formulate this process as follows:

$$S'^{(l-1)} = \text{SN}(S^{(l-1)}), \tag{3}$$

$$Q = \text{LBS}(S'^{(l-1)}), \ K = \text{LBS}(S'^{(l-1)}), \ V = \text{LBS}(S'^{(l-1)}). \tag{4}$$

Here, $Q, K, V \in \mathbb{R}^{T \times C \times N}$ are all in spike form, N represents the number of points and C represents the number of channels. The spiking vector attention mechanism we employ performs distinct vector computations between $Q$ and $K$, which are then converted into spike values through a Spiking Neural Network (SN). Subsequently, a Spike Point Masker (SPM) is used to define sparse attention computation, yielding the attention feature $E_{\text{va}}$. Compared to traditional attention computation in Artificial Neural Network (ANN), which requires energy-intensive multiply-accumulate (MAC) operations between floating-point numbers, the spiking design demonstrates superior energy efficiency. Next, when combined with the spiking offset attention layer, the offset (difference) between the attention feature and the input feature $S'^{(l-1)}$ is computed via element-wise subtraction. This offset is then reconverted into a spike sequence through the SN layer, followed by Linear and batch normalization operations. Finally, it is added to the input feature $S'^{(l-1)}$ to provide the output feature $S_1^{(l)}$ for the spiking-based attention module. This process can be expressed as:

$$E_{va} = \text{SPM}(\text{SN}(\tau(Q,K)), V), \tau(Q,K) = Q - K, \tag{5,6}$$

$$S_1^{(l)} = \text{BN}(\text{Linear}(\text{SN}(S'^{(l-1)} - E_{va}))) + S'^{(l-1)}. \tag{7}$$

Where $\tau$ denotes the computational form of spiking vector attention, implemented as channel subtraction in this paper. SPM represents the Spike Point Masker (detailed in the next subsection), and $E_{\text{va}}$ is the resulting spiking vector feature. Compared to traditional attention mechanisms in Artificial Neural Network (ANN), our spiking attention computation is entirely based on spike-based representations of query, key, and value vectors. By focusing on critical information, aggregating relevant features, and discarding irrelevant ones, our method significantly improves computational efficiency and energy performance.

### 3.2.2 SPIKE POINT MASKER

Traditional 3D point cloud attention uses softmax and scaling to balance attention weights for numerical stability. However, Spiking Neural Network (SNN) output discrete spike sequences, making softmax incompatible. QKformer Zhou et al. (2024a) addresses this by a linear attention based on $Q \odot K$, but it relies on the strong spatial correlation of adjacent pixels in 2D images. Since 3D point clouds are sparse and irregularly distributed, using only $Q \odot K$ can misdirect attention to invalid points (spike=0) and obscure valid features. To tackle this, we propose a novel Spike Point Masker (SPM) module (Figure 3) with two design variants. The first structure is as follows:

$$S_C = \text{SN}\left(\sum_{i=0}^{N} V_{i,j}\right), \quad S'_C = S_C \otimes \text{SN}(\tau(Q,K)). \tag{8,9}$$

Here, $S_C$ is a $C \times 1$ channel attention vector that models the binary importance of different channels. $S_C$ is a spike-form vector obtained by summing the rows of the spike matrix $V$ and then processing it through spike neurons. $\otimes$ denotes the Hadamard product between spike tensors. Since spike values are in $\{0, 1\}$, the product range remains unchanged, making it equivalent to a masking operation. We apply the spike-form Channel attention vector $S_C$ to the spike attention matrix using the Channel Mask to compute the output $S'_C$. Similarly, the second structure can be expressed as follows:

$$S_N = \text{SN}\left(\sum_{j=0}^{C} V_{i,j}\right), \quad S'_N = S_N \otimes \text{SN}(\tau(Q,K)). \tag{10,11}$$

Figure 3: The design of the Spike Point Masker (SPM). The left side represents a spiking Channel Masker that performs Row summation on channels, and the right side represents a spiking Npoint Masker that performs Column summation on Npoints.

Where $S_N$ is an $1 \times N$ Npoint attention vector that models the binary importance of different points. $S_N$ is a spike-form vector obtained by columns-wise summation of the spike matrix $V$ followed by processing through a spiking neuron. Subsequently, the output $S'_N$ is obtained by performing a Column-wise masking operation (Npoint Mask) between $S_N$ and the spike attention matrix. The proposed Spike Point Masker module fully leverages the spiking characteristics of SNN, enabling efficient processing of point cloud data and focusing on key information extraction without relying on softmax and scaling operations.

## 3.3 SPIKING POINT CLOUD CLASSIFICATION

The verview of SpikeNet for 3D point cloud classification is shown in the upper part of Figure 4. It consists of three parts: Embedding, Spiking Encoder, and Classification Head. The first part is the ANN-based Embedding. It takes a point cloud $P \in \mathbb{R}^{N \times 3}$ as input, where each of the $N$ points is described by a 3D coordinate, and outputs a corresponding feature set of $d_{1st}$ dimensions. Then, the learned feature set $F_{1st}$ is input into the second part, the Spiking Encoder. The second part is divided into 4 Stages in total for point cloud hierarchical spike feature enhancement. Each Stage can be described by the following formula:

$$\text{Stage} = \text{SVMT}(\text{AMP}(\text{DSSR}(\text{GAM}(F_{1st})))). \tag{12}$$

Here, the Geometric Affine Module(GAM) was proposed in PointMLPMa et al. (2022).Similar to the sampling and grouping in PointNet++Qi et al. (2017b), the GAM enhances the expressive ability of local features by performing affine transformations on local geometric features.The extracted lightweight local point features are input into our proposed DSSR module. With the help of the residual structure, the obtained deep features are processed by Adaptive-Max-Pooling (AMP) to highlight the key information in the local area of the point cloud. Then, these features are input into the sparse SVMT module to extract the representative features of deep aggregation. The SVMT combines the temporal characteristics of Spike and the advantages of Transformer in modeling long-range dependencies and adaptively fusing features, enabling the model to better understand the overall structure and semantic information of point cloud data. Finally, the high-precision float linear layer in the classification head (CH) predicts the final classification scores of $N_c$ object categories (such as tables, beds, etc.). The specific description can be expressed by the following formula:

$$F_{1st} = \text{Embedding}(P), \qquad F_{1st} \in \mathbb{R}^{N \times C}, \tag{13}$$

$$F_{s1} = \text{Stage}_1(F_{1st}), \qquad F_{s1} \in \mathbb{R}^{\frac{N}{2} \times 2C}, \tag{14}$$

$$F_{si} = \text{Stage}_i(F_{s(i-1)}), \qquad F_{si} \in \mathbb{R}^{\frac{N}{2^i} \times 2^i C}, \quad i = 2, 3, 4, \tag{15}$$

$$Category = \text{CH}(F_{s4}). \tag{16}$$

Here, the Embedding consists of Conv-BN-ReLU connected in sequence, providing the subsequent spiking network layers with inputs that are rich in feature information and have a unified dimension. The Spiking Encoder is responsible for important spiking feature encoding and information processing tasks. The input of each stage in it is obtained by doubling the dimension and halving the number of points of the output from the previous stage, gradually expanding the receptive field to improve performance. The Classification Head is composed of a simple Linear layer, which maps

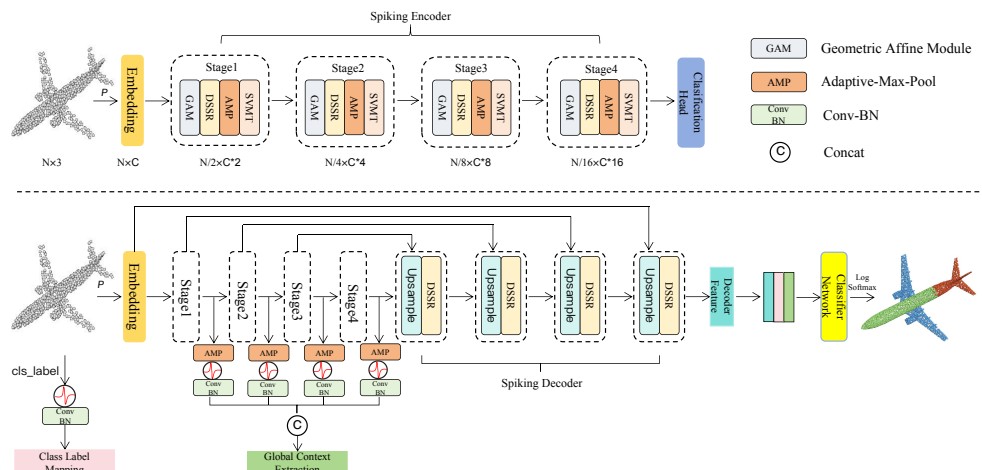

Figure 4: SpikeNet overview. First row shows our point cloud shape classification network, and second row shows our point cloud part segmentation network.

high dimensional feature vectors to the corresponding classification category space, thereby outputting the probability values of each sample belonging to different categories and completing the classification task of the input data.

## 3.4 SPIKING POINT CLOUD SEGMENTATION

For the part segmentation task, as shown in the lower part of Figure 4, we adopted the framework proposed in PointNet++Qi et al. (2017b). We utilized a hierarchical propagation strategy based on distance interpolation and cross-layer skip connections, and replaced the backbone network with our SpikeNet. The difference lies in the fact that our Upsample is composed of SN-Conv-BN connections, and the MLP in the cross-layer connections is also replaced by our proposed DSSR module. Subsequently, we performed spike feature transformation on the input category labels using SN-Conv-BN connections. We also applied an Adaptive-Max-Pooling (AMP) operation to the features extracted from each stage of the spike encoder, followed by SN-Conv-BN connections, to concatenate the feature vectors from all stages in the channel dimension, obtaining the final global context features. Finally, we concatenated these features with those obtained from the backbone network. Through a classifier composed of a simple linear layer, we obtained the scores for each point belonging to different categories, thereby achieving part segmentation of the point cloud.

## 4 EXPERIMENT

We evaluate the performance of the SpikeNet classification network on two public benchmark datasets, ModelNet40Wu et al. (2015) and ScanObjectNNUy et al. (2019), and the performance of the SpikeNet segmentation network on the 3D part segmentation benchmark dataset, ShapeNet-PartYi et al. (2016). The experiments are carried out on a server equipped with an AMD Ryzen Threadripper 3960X 3.80GHz 24-core processor and an NVIDIA A6000 GPU. Our implementation is based on PyTorch and SpikingJellyFang et al. (2023). For spiking neurons, we set the time latency $T$ to a default value of 3. For the specific introduction of the datasets and the detailed experimental settings, please refer to the appendix.

### 4.1 COMPARISONS WITH STATE-OF-THE-ART METHODS

In this section, we conduct a rigorous comparison the performance of SpikeNet with some point cloud analysis networks based on ANN as well as point cloud understanding architectures based on SNN. It should be noted that there are two different variants of SpikeNet. The SpikeNet-N network is constructed based on the Npoint Masker, while the SpikeNet-C network is built based on the Channel Masker. In terms of evaluation metrics, We employ overall accuracy (OA) and mean

| Methods | Year | Type | Para. | Flops | ModelNet40 | | | ScanObjectNN | | |
|---|---|---|---|---|---|---|---|---|---|---|
| | | | | | OA | mAcc | Energy | OA | mAcc | Energy |
| PointNeXt | 22'NIPS | ANN | 1.4 | 3.6 | 94.0 | 90.8 | 16.6 | 87.7 | - | 16.6 |
| Point-BERT | 22'CVPR | ANN | 0.8 | 1.0 | 93.2 | - | 22.1 | 83.1 | - | 22.1 |
| PointMLP | 22'ICLR | ANN | 12.6 | 12.8 | 94.1 | 91.5 | 59.0 | 85.4 | 83.9 | 59.0 |
| PointNN | 23'CVPR | ANN | 0.8 | 1.0 | 93.8 | - | 4.6 | 87.1 | - | 4.6 |
| PointMamba | 24'NIPS | ANN | 12.3 | 3.6 | 92.4 | - | 16.6 | 82.5 | - | 16.6 |
| PoinTramba | 24'ICLR | ANN | 19.5 | 5.7 | 92.9 | - | 26.2 | 89.1 | - | 26.2 |
| PCM | 25'AAAI | ANN | 34.2 | 45.0 | 93.4 | - | 207.0 | 88.1 | - | 207.0 |
| SpikePointNet | 23'ICCV | SNN | 3.5 | 1.6 | 88.6 | - | 1.4 | 69.2 | - | 1.4 |
| P2SResLNet | 24'AAAI | SNN | 6.2 | 3.3 | 90.6 | 89.2 | 3.0 | 81.0 | 79.3 | 3.1 |
| Spike PointNet | 24'NIPS | SNN | 3.5 | 0.4 | 88.2 | 86.7 | 0.4 | 66.4 | 60.4 | 0.4 |
| E-3DSNN | 25'AAAI | SNN | 3.3 | - | 91.7 | - | - | - | - | - |
| SPT | 25'AAAI | SNN | - | 14.0 | 91.4 | 89.4 | 13.3 | 78.0 | 75.9 | - |
| **SpikeNet-N (ours)** | - | SNN | 10.5 | 1.9 | **93.2 (↑ 1.5)** | **90.4 (↑ 1.0)** | 1.7 | **85.0 (↑ 4.0)** | **83.2 (↑ 3.9)** | 1.8 |
| **SpikeNet-C (ours)** | - | SNN | 10.5 | 1.9 | **93.4 (↑ 1.7)** | **90.9 (↑ 1.5)** | 1.7 | **85.5 (↑ 4.5)** | **83.3 (↑ 4.0)** | 1.8 |

Table 1: Comparison with existing methods on two datasets. '-' means the model did not provide results, and bold denotes important results. Our method gives the best Energy-Accuracy balance even though we did not use the voting strategy for computing our method's results. OA and mAcc are in %, the units of Energy is millijoule (mJ), Parameters are in millions and Flops are in Giga.

accuracy per class (mAcc) as the evaluation metrics for point cloud classification and employed category mIoU(Cat.mIoU) and Instance mIoU(Ins.mIoU) for point cloud part segmentation.

**Classification on ModelNet40.** Compared to traditional classifiers based on ANN, models based on SNN typically exhibit lower recognition accuracy due to the spike-based nature of their features. However, there is significant room for performance improvement, and they are expected to receive more attention in the future. We first conduct point cloud classification experiments on the ModelNet40Wu et al. (2015) dataset. As shown in Table 1, from the perspective of the SNN-based design concept, the accuracy of Spiking PointNet Ren et al. (2024) is only 88.61%, that of P2SResLNet Wu et al. (2024) is 90.6%, and the accuracy of the current SOTA model E-3DSNNQiu et al. (2025) is only 91.7%. Our SpikeNet achieves an optimal accuracy of 93.4%, even surpassing many ANN-based point cloud classification networks. Meanwhile, our method demonstrates excellent performance in energy consumption. Compared with the SNN-based SPT Wu et al. (2025) model, the energy consumption of our method is reduced by 9 times; compared with the ANN-based PointMLP Ma et al. (2022) model, it is reduced by 34 times.

**Classification on ScanObjectNN.** ScanObjectNNUy et al. (2019) is a dataset based on real-world scans, featuring varying data loss and noise pollution. Such complexity challenges many ANN-based 3D classifiers to achieve high accuracy. Traditional Transformer architectures struggle with incomplete/occluded point clouds and lack performance validation in such scenarios. Our SpikeNet model based on SNN shows exceptional performance. As Table 1 shows, it achieves 85.5% accuracy on ScanObjectNN, surpassing classic ANN networks like PointMLPMa et al. (2022). Notably, our method reduces Flops by 7 times and energy consumption by 33 times compared to PointMLP. It also outperforms SOTA SNN-based classifiers like P2SResLNet Wu et al. (2024) (81.0% accuracy), cutting Flops and energy by 40%. SpikeNet's performance on ScanObjectNN proves its breakthrough effectiveness in real complex scenarios.

**Part Segmentation on ShapeNetPart.** Point cloud segmentation is a highly challenging task, aiming to divide a 3D model into multiple meaningful parts. In 3D point cloud tasks designed based on Spiking Neural Network (SNN), our SpikeNet has achieved 3D shape part segmentation. We conducted tests on the ShapeNetPartYi et al. (2016) benchmark dataset and compared our method with several

| Methods | Year | Type | Cat.mIoU | Ins.mIoU |
|---|---|---|---|---|
| PointNet++ | 17'NIPS | ANN | 81.9 | 85.1 |
| APES | 23'CVPR | ANN | 83.1 | 85.6 |
| PointMamba | 24'NIPS | ANN | 84.4 | 86.0 |
| PCM | 25'AAAI | ANN | 85.6 | 87.1 |
| SPT | 25'AAAI | SNN | 81.3 | 82.9 |
| SpikeNet-N | - | SNN | **83.3(↑ 2.0)** | **85.1(↑ 2.2)** |
| SpikeNet-C | - | SNN | **83.9(↑ 2.6)** | **85.4(↑ 2.5)** |

Table 2: Segmentation results on ShapeNet Part.

recent studies based on Artificial Neural Network (ANN). As shown in Table 2, our SpikeNet demonstrates superior performance within the SNN domain, achieving 83.9% Cat. mIoU and 85.4% Ins. mIoU, surpassing SPTWu et al. (2025) by 2.6% and 2.5%, respectively. our SNN-based method

| Method | Type | Type | mIoU | ceiling | floor | wall | beam | column | window | door | table | chair | sofa | bookcase | board | clutter | E |
|--------|------|------|------|---------|-------|------|------|--------|--------|------|-------|-------|------|----------|-------|---------|---|
| PointNet | 17'CVPR | ANN | 41.1 | 88.8 | 97.3 | 69.8 | 0.0 | 3.9 | 46.3 | 10.8 | 59.0 | 52.6 | 5.9 | 40.3 | 26.4 | 33.2 | 5.5 |
| PointNet++ | 17'NIPS | ANN | 53.5 | 89.4 | 97.7 | 75.4 | 0.0 | 1.8 | 58.3 | 19.5 | 79.0 | 69.2 | 59.1 | 46.2 | 58.7 | 41.6 | 5.5 |
| PointCNN | 18'NIPS | ANN | 57.3 | 92.3 | 98.2 | 79.4 | 0.0 | 17.6 | 22.8 | 62.1 | 74.4 | 80.6 | 31.7 | 66.7 | 62.1 | 56.7 | 324.5 |
| PointNeXt | 22'NIPS | ANN | 70.5 | 94.2 | 98.5 | 84.4 | 0.0 | 37.7 | 59.3 | 74.0 | 83.1 | 91.6 | 77.4 | 77.2 | 78.8 | 60.6 | - |
| PCM | 25'AAAI | ANN | 63.4 | 93.3 | 96.7 | 80.6 | 0.0 | 35.9 | 57.7 | 60.0 | 74.0 | 87.6 | 50.1 | 69.4 | 63.5 | 55.9 | - |
| PointRWKV | 25'AAAI | ANN | 70.5 | 94.2 | 98.3 | 86.5 | 0.0 | 38.6 | 64.5 | 76.2 | 88.2 | 89.3 | 65.2 | 75.6 | 78.2 | 61.3 | - |
| PTv1 | 21'ICCV | ANN | 70.4 | 94.0 | 98.5 | 86.3 | 0.0 | 38.0 | 63.4 | 74.3 | 89.1 | 82.4 | 74.3 | 80.2 | 76.0 | 59.3 | 76.8 |
| PTv2 | 22'NIPS | ANN | 71.6 | 93.0 | 98.1 | 86.7 | 0.0 | 48.0 | 62.4 | 76.1 | 88.3 | 87.6 | 77.1 | 79.2 | 77.5 | 59.8 | 400.1 |
| PTv3 | 24'CVPR | ANN | 73.6 | 92.4 | 98.3 | 86.6 | 0.0 | 55.8 | 63.7 | 77.1 | 83.8 | 93.3 | 79.1 | 79.4 | 85.4 | 61.7 | 687.7 |
| E-3DSNN | 25'AAAI | SNN | 67.4 | 95.3 | 98.5 | 82.3 | 0.0 | 28.0 | 55.8 | 71.5 | 81.2 | 89.8 | 69.2 | 76.4 | 67.0 | 61.6 | 14.4 |
| SpikeNet-N | - | SNN | **68.4** | 89.9 | 91.5 | 83.0 | 0.0 | 47.0 | 60.1 | 74.6 | 79.7 | 86.0 | 69.6 | 71.5 | 80.2 | 57.0 | **10.2** |
| SpikeNet-C | - | SNN | **68.9** | 90.1 | 92.2 | 83.5 | 0.0 | 47.6 | 60.5 | 75.2 | 80.1 | 86.3 | 70.3 | 72.4 | 80.6 | 57.3 | **10.2** |

Table 3: Semantic Segmentation Results on S3DIS Dataset. E represents energy consumption. The unit is mJ.

also outperforms the classic ANN-based PointNet++. Our work is groundbreaking, achieving amazing point cloud segmentation with high performance and low energy consumption.

**Semantic segmentation on S3DIS.** We conducted semantic segmentation experiments on the S3DIS dataset Armeni et al. (2016), and Table 3 presents the performance and energy efficiency of SpikeNet compared with other ANN and SNN methods. Among ANN methods, PTv3 achieves the best performance (mIoU 73.6%) but has an very high energy consumption of 687.7 mJ. Among SNN methods, our SpikeNet-N (68.4%) and SpikeNet-C (68.9%) significantly outperform E-3DSNN (67.4%). Both have a low energy consumption of only 10.2 mJ, lower than E-3DSNN (14.4 mJ) and most ANN methods, achieving a balance between accuracy and energy efficiency. These results verify the advantages of SNN model like SpikeNet: they effectively capture scene semantics, retain low-energy consumption characteristics, and have potential for practical applications.

| Method | Type | Input | Val | Test |
|--------|------|-------|-----|------|
| SPVNAS | ANN | point | 64.7 | 66.4 |
| Cylinder3D | ANN | point | 64.3 | 67.8 |
| AF2S3Net | ANN | point | 74.2 | 70.8 |
| PTv2 | ANN | point | 70.3 | 72.6 |
| PTv3 | ANN | point | 72.3 | 75.5 |
| E-3DSNN | SNN | voxel | 63.2 | 69.4 |
| SpikeNet-N(our) | SNN | point | 64.6 | 69.8 |
| SpikeNet-C(our) | SNN | point | 65.1 | 70.2 |

Table 4: Scene segmentation results on Semantic KITTI Dataset.

| Method | Type | Input | Val | Test |
|--------|------|-------|-----|------|
| PointNeXt | ANN | point | 71.5 | 71.2 |
| PointMetaBase | ANN | point | 72.8 | 71.4 |
| MinkUNet | ANN | voxel | 72.2 | 73.6 |
| PTv2 | ANN | point | 75.4 | 75.2 |
| PTv3 | ANN | point | 77.5 | 77.9 |
| E-3DSNN | SNN | voxel | 68.2 | 69.5 |
| SpikeNet-N(our) | SNN | point | 69.4 | 70.8 |
| SpikeNet-C(our) | SNN | point | 69.7 | 71.2 |

Table 5: Semantic segmentation results on ScanNet V2 Dataset.

**Scene segmentation on Semantic KITTI.** We compared SpikeNet with other SNN-based and ANN-based methods on the Semantic KITTI Behley et al. (2019), using mIoU (validation/test set) as the metric. As shown in Table 4, among SNN methods, our SpikeNet-N (64.6%/69.8%) and SpikeNet-C (65.1%/70.2%) outperform E-3DSNN (63.2%/69.4%). Due to the mature development, refined optimization of ANN and no constraints from spiking encoding, the mIoU of SpikeNet is slightly lower than that of advanced ANN methods such as PTv3 (75.5% on test set). However, as a point cloud SNN model, SpikeNet maintains competitiveness in outdoor 3D segmentation while inheriting the low-energy consumption characteristic, demonstrating unique practical value.

**Semantic segmentation on ScanNet V2.** We compared SpikeNet with representative ANN-based and SNN-based methods on the ScanNet V2Dai et al. (2017). The evaluation metric is mean Intersection over Union (mIoU, validation set/test set). As shown in Table 5, among SNN methods, our proposed SpikeNet-N (69.4%/70.8%) and SpikeNet-C (69.7%/71.2%) perform better than E-3DSNN (68.2%/69.5% on test set), and are only slightly lower than PTv3 (77.5%/77.9%), an advanced ANN method. ANN have been optimized in structure and parameters for a long time, so their accuracy is usually higher than SNN. However, SpikeNet still maintains certain competitiveness in indoor 3D semantic segmentation tasks, which is a promising direction for point cloud processing.

### 4.2 ABLATION STUDIES

We use ModelNet40 as baseline dataset for ablation study to determine final network architecture.

**Effect of Embedding Dimension.** Most network-based method experiments show that better performance can be achieved when using a larger embedding dimension. In our experiments, the em-

bedding dimension of 64 is set as the default value. We also report the results with embedding dimensions of 32 and 96 in Table 6.

**Effect of Spiking Vector Attention Operator $\tau$.** Self-attention mechanism operators can generally be divided into two types: scalar attention and vector attention. The former conducts feature-level similarity estimation, while the latter performs channel-level similarity estimation. Each operator can be defined as Summation $(+)$, Subtraction $(-)$, Hadamard product $(\odot)$, and Division $(\div)$. We apply all the above-mentioned operators to our SpikeNet framework to evaluate their performance. As can be seen from the results in Table 7, the vector attention in the form of subtraction has higher accuracy and is more suitable for our point cloud classification framework.

**Effect of Spike Point Masker.** The Spike Point Masker module we proposed has two different designs: Npoint Masker and Channel Masker. Table 8 presents the evaluation results on the Model-Net40 test set. Experiments show that the design of Channel Masker can achieve better performance. This is because the Channel Masker better aligns with the distribution characteristics of point cloud features in the channel dimension, and can accurately filter key channel information (such as semantic attributes like shape and curvature), whereas the Npoint Masker tends to cause the loss of key points in point clouds, leading to performance degradation. Therefore, in subsequent experiments, we adopt the Channel Masker structure as the final framework.

| Embedding Dimension | 32 | 64 | 128 |
|---|---|---|---|
| OA (%) | 92.7 | **93.4** | 92.9 |

Table 6: Effect of embedding dimensions

| $\tau$ | + | - | $\odot$ | $\div$ |
|---|---|---|---|---|
| OA (%) | 93.2 | **93.4** | 93.0 | 92.7 |

Table 7: Effect of $\tau$ in Spiking Vector Attention

| Spike Point Masker | OA (%) | mAcc (%) |
|---|---|---|
| Npoint Masker | 93.2 | 90.4 |
| Channel Masker | **93.4** | **90.9** |

Table 8: Effect of Spike Point Masker

| $T$ | OA (%) | mAcc (%) | $T$ | OA (%) | mAcc (%) |
|---|---|---|---|---|---|
| 1 | 92.8 | 89.8 | 3 | **93.4** | **90.9** |
| 2 | 93.1 | 90.2 | 4 | 93.2 | 90.4 |

Table 9: Analysis on time latency

**Analysis on Time Latency.** Time delay is crucial in Spiking Neural Networks as it determines the model's computational granularity in the temporal dimension and feature-extraction sequentiality. We aim to calibrate it for an optimal balance between classification accuracy and efficiency. Based on the aforementioned ablation experiments, we set $T = 1, 2, 3, 4$. As Table 9 shows, when $T = 3$, the SpikeNet strikes a good balance, achieving a $93.4\%$ accuracy. At this delay, SpikeNet's structure is optimally configured for spike-signal propagation, enhancing feature integration and classification accuracy. Increasing $T = 4$ offers a wider time range for signal propagation and processing but also significantly raises computational demands. Moreover, it reduce salience of temporal features and affect prediction accuracy. Therefore, we set the default value of the time delay $T$ to 3.

**Analysis on Efficiency.** We compared training and inference overhead on ModelNet40, focusing on two metrics: Latency and Peak Memory. As shown in Table 10, when the number of sampled points increases from 512 to 1024, the latency and memory usage of SNN-based SPT series models rise significantly. Mamba3D, which is based on ANN, also maintains a high resource consumption level. In contrast, our proposed SpikeNet reduces training latency to 321 ms and memory usage to 11.3 GB; in the inference phase, it only requires 185 ms and 6.1 GB. These results are better than existing comparison methods, verifying that SpikeNet balances time efficiency and hardware friendliness while maintaining accuracy.

**Analysis on Spike Encoder.** Using different modules in each stage of the Spike Encoder yields different results. We fixed the GAM and AMP modules and carried out multiple ablation experiments on the newly proposed DSSR and SVMT modules. In these experiments, we not only altered the implementation positions of these two modules but also tested the functions of each individual module. The results, as shown in Table 11, indicate that the adopted GAM-DSSR-AMP-SVMT structure demonstrates the optimal performance, suggesting that both the DSSR and SVMT modules play pivotal roles. Note that MLP* represents Conv-BN-ReLU based on ANN, and MLP represents SN-Conv-BN based on SNN.

| Methods | Training | | Inference | |
|---|---|---|---|---|
| | **Latency** | **Memory** | **Latency** | **Memory** |
| SPT-512 | 326ms | 9.7G | 191ms | 5.2G |
| SPT-768 | 385ms | 12.5G | 201ms | 7.3G |
| SPT-1024 | 431ms | 15.2G | 227ms | 9.5G |
| Mamba3D | 433ms | 14.9G | 256ms | 7.8G |
| SpikeNet(our) | **321ms** | **11.3G** | **185ms** | **6.1G** |

Table 10: Model Efficiency on ModelNet40.

| Spike Encoder | OA (%) | mAcc (%) |
|---|---|---|
| GAM-MLP*-AMP-MLP* | 92.7 | 90.1 |
| GAM-MLP-AMP-MLP | 90.1 | 86.7 |
| GAM-DSSR-AMP-DSSR | 92.2 | 89.3 |
| GAM-SVMT-AMP-SVMT | 92.0 | 88.9 |
| GAM-SVMT-AMP-DSSR | 92.7 | 89.6 |
| GAM-DSSR-AMP-SVMT | **93.4** | **90.9** |

Table 11: Analysis of Spiking Encoder

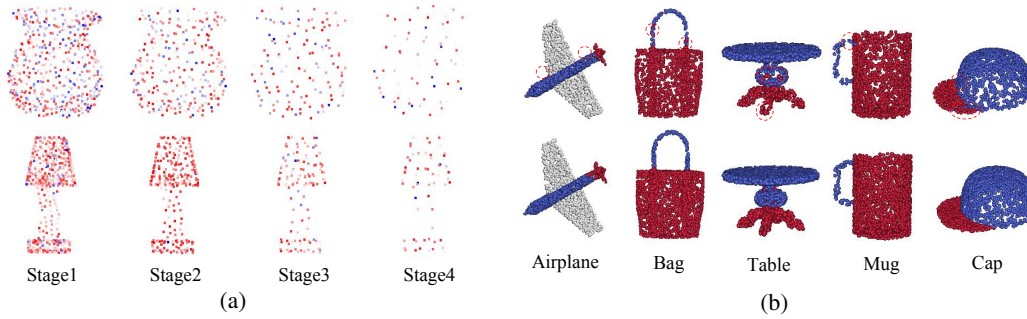

Figure 5: (a) shows the visualize the features learned at different stages in the Spike Encoder of SpikeNet-C on ModelNet 40. (b) shows the Part segmentation results on ShapeNetPart. Top line is our prediction and bottom line is ground truth.

### 4.3 FEATURE VISUALIZATION

To explore the feature learning mechanism of SpikeNet, we conduct feature visualization analysis at different stages for the Spike Encoder of the SpikeNet-C classification network. For the input point cloud data, we first average the spike features of 3D points in temporal and channel dimensions, then perform pseudo-color encoding based on the spiking activity rate (Figure5 (a)). Cool colors (blue hues) represent a low activity rate (close to zero), while warm colors (red hues) denote a high activity rate. The visualization results show that as the network layer depth increases, the number of point cloud sampling points decreases gradually, points with a high firing rate have significant feature retention advantages; and the hierarchical feature representativeness enhances step by step.

We compare the predicted results with the ground truth annotations in the segmentation task of parts, as shown in Figure 5 (b). Although there remains room for improvement in edge sharpness (e.g., aircraft tail contours, cup rim edges) and component integrity (e.g., bag handles, tabletop central regions, hat crown structures) as highlighted in red boxes, our method provides an innovative solution with remarkable energy efficiency advantages for 3D point cloud segmentation tasks.

## 5 CONCLUSION

We introduce SpikeNet, an innovative spiking neural network model designed for point cloud processing. SpikeNet effectively addresses the high energy consumption issues of traditional point cloud models based on Artificial Neural Network (ANN). By combining the stability of residual structures with the powerful feature extraction capabilities of transformers, it improves the robustness of Spiking Neural Network (SNN) during training, enabling more efficient and accurate feature capture in point cloud data. This provides a novel and promising solution for point cloud analysis. Extensive experiments show that SpikeNet exhibits outstanding performance, reaching a leading level in the field. It rivals traditional ANN-based models in performance while achieving remarkable energy-saving advantages. In future work, we plan to expand the application of SpikeNet to more point cloud tasks and further explore the potential of SNN in 3D data processing.

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

## A  SPIKING NEURON MODEL

Spiking neuron is the fundamental unit of SNN, we choose Integrate-and-Fire (IF) model as the spiking neuron in the proposed SpikeNet. The dynamics of a IF neuron can be formulated as follows:

$$u_t = V_{t-1} + I_t \tag{1}$$
$$S_t = \Theta(u_t - V_{th}) \tag{2}$$
$$V_t = u_t(1 - S_t) + V_{reset}S_t \tag{3}$$

Where $V_{t-1}$ represents the membrane potential of the neuron at time step $t - 1$, which undergoes a spike trigger evaluation. $I_t$ represents the input current at time step $t$. $u_t$ denotes the membrane potential of the neuron after incorporating neuron dynamics at time step $t$. In Equation (2), $\Theta(\cdot)$ symbolizes the spike trigger evaluation at time step $t$. When the membrane potential $u_t$ exceeds the firing threshold $V_{th}$, the neuron emits a spike, setting $S_t = 1$; otherwise, $S_t = 0$. In Equation (3), $V_t$ represents the membrane potential of the neuron at time step $t$ after the spike evaluation. If no spike is produced, $V_t$ is equivalent to $u_t$; otherwise, it is reset to the potential $V_{reset}$.

## B  PROOF OF SNN STABILITY IN TRAINING WITH THE DSSR MODULE

The core of proving the gradient stability of SNN training with the Dynamic Sparse Spiking Residual (DSSR) module lies in constraining the gradient norm to avoid explosion or vanishing. Here, we clarify the role of the spiking surrogate gradient and dynamic sparsity in stabilising the training.

### B.1  PROOF OF SPIKE SURROGATE GRADIENT STABILITY

Define the loss gradient of layer $l$ as $\nabla\mathcal{L}^{(l)} = \partial\mathcal{L}/\partial Y^{(l)}$. From equation 1 in the main paper, the derivative of the DSSR output $Y^{(l)}$ with respect to the pre-activation $Y_1^{(l)}$ can be written as

$$\frac{\partial Y^{(l)}}{\partial Y_1^{(l)}} = \frac{\partial}{\partial Y_1^{(l)}}\Big[\mathrm{BN}\big(\mathrm{Conv}(\mathrm{SN}(Y_1^{(l)}))\big)\Big] + I, \tag{4}$$

where $I$ is identity matrix. The derivatives of convolution (Conv) and batch normalization (BN) operations are bounded, with their bound denoted as $K$. The gradient of the spiking neuron $\mathrm{SN}(\cdot)$ is approximated by the ATan function (with the specific form equation 26), and this approximate derivative satisfies the boundedness of $\leq \alpha/2$. Therefore, the norm of the derivative of $Y^{(l)}$ with respect to $Y_1^{(l)}$ satisfies

$$\left\|\frac{\partial Y^{(l)}}{\partial Y_1^{(l)}}\right\| \leq K \cdot \frac{\alpha}{2} + 1. \tag{5}$$

Based on equation 2 , the derivative of $Y_1^{(l)}$ with respect to $X_1^{(l)}$ also has similar boundedness. By induction, it can be shown that the deep gradient $\nabla\mathcal{L}^{(l)}$ and the shallow gradient $\nabla\mathcal{L}^{(l-1)}$ satisfy

$$\|\nabla\mathcal{L}^{(l)}\| \leq L \cdot \|\nabla\mathcal{L}^{(l-1)}\|, \quad L = \left(K \cdot \frac{\alpha}{2} + 1\right)^2. \tag{6,7}$$

When $L < 1$ by adjusting $\alpha$ and $K$, the gradient norm decays with the number of network layers but does not vanish, thereby ensuring the stability of SNN training.

### B.2  DYNAMIC SPARSITY AND EFFECTIVE GRADIENT BOUND

The DSSR module includes a spiking neuron gate, which generates discrete spikes at a rate $p$ (the fraction of time steps when SN outputs a spike). Because convolution is only executed on those activations where a spike occurs, the average gradient flowing through the convolutional branch is scaled by $p$. Formally, if the full-activity gradient bound is $K$, then the effective bound becomes $pK$. Consequently the gradient amplification factor is reduced to

$$L_{\mathrm{eff}} \approx \left(pK\frac{\alpha}{2} + 1\right)^2, \tag{8}$$

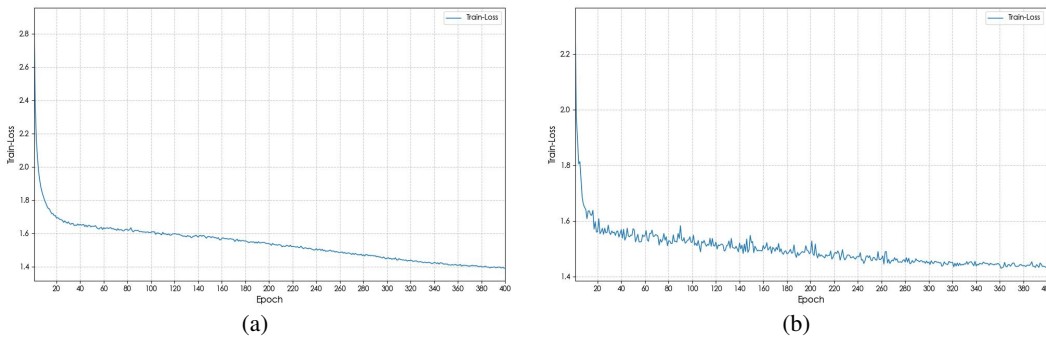

(a)                                        (b)

Figure 6: (a) Training loss curve of SpikeNet with DSSR module. (b) Training loss curve of SpikeNet without DSSR module.

which can be less than one even when $K\alpha/2 > 1$ provided $p$ is sufficiently small. This observation explains why the dynamic sparsity of spiking neurons is beneficial: it not only reduces computation but also effectively scales down the back-propagated gradients. These theoretical analyses on gradient stability and effective gradient scaling are further validated by the training loss curves of models with and without the DSSR module, as shown in Figure. 6, where the DSSR-equipped model exhibits more stable convergence and avoids gradient explosion/vanishing issues.

## C  MATHEMATICAL ANALYSIS OF THE SVMT MODULE

The Spiking Vector Mask Transformer (SVMT) is a novel component in the SpikeNet architecture for point cloud classification and segmentation. It integrates spiking neurons with a Transformer–style attention mechanism and introduces a dynamic masking strategy to reduce computation on sparse point clouds.

### C.1  MODEL DEFINITION

Let $S^{(\ell-1)} \in \{0, 1\}^{T \times C \times N}$ denote the binary spike tensor at layer $\ell - 1$ where $T$ is the number of time steps, $C$ is the channel dimension and $N$ is the number of points. A linear layer, batch normalisation and spiking neuron are applied sequentially to produce the query $Q$, key $K$ and value $V$ tensors:

$$Q = \text{SN}(\text{BN}(\text{Linear}(S^{(\ell-1)}))), K = \text{SN}(\text{BN}(\text{Linear}(S^{(\ell-1)}))), V = \text{SN}(\text{BN}(\text{Linear}(S^{(\ell-1)}))), \quad (9)$$

where SN denotes the spiking neuron activation that outputs binary spikes.

For each time index $t$, channel $c$ and point $i$, an element–wise vector difference is computed and passed through a spiking neuron to produce the binary attention indicator:

$$A_{t,c,i} = \text{SN}\big(\tau(Q_{t,c,i}, K_{t,c,i})\big) \in \{0, 1\}, \quad (10)$$

where $\tau$ is a vector operator ($\tau = Q - K$ in our work). An entry $A_{t,c,i} = 1$ indicates that the query and key differ sufficiently at that spatiotemporal location, while $A_{t,c,i} = 0$ means no attention is generated.

### C.2  IMPORTANCE OF THE Q–K DIFFERENCE FOR POINT CLOUDS

In our proposed SVMT, the operator $\tau$ is implemented as a simple difference $\tau(q, k) = q - k$. This design is specifically tailored for point cloud processing for several reasons. First, in point cloud processing, each point $i$ carries a set of features across channels (e.g. spatial coordinates, normals or learned embeddings). Computing a pairwise dot product between features of different points, as in conventional self-attention, is unnecessary when the goal is to detect local variations within a point's feature vector. By taking the difference of the query and key in each channel,

$$\tau(Q_{t,c,i}, K_{t,c,i}) = Q_{t,c,i} - K_{t,c,i}, \quad (11)$$

SVMT measures how strongly the feature at channel $c$ changes between the query and key streams. Passing this difference through a spiking neuron yields a binary gate $A_{t,c,i}$ that indicates whether or not the variation exceeds a threshold. This channel-wise gating emphasizes local geometric cues: channels that encode similar information (e.g. neighboring points with nearly identical coordinates) produce small differences and therefore do not fire, whereas channels that capture distinctive local structure trigger a spike. The resulting attention therefore focuses computational resources on structurally informative channels rather than all channels equally.

This channel-wise difference is particularly important for irregular point clouds, where each point is independent and there is no implicit neighborhood ordering. In contrast to vision transformers for images, there is no natural spatial grid for computing convolutional filters. Our SVMT's per-channel difference therefore acts as a local operator that highlights salient features without resorting to expensive pairwise comparisons across all points. Because Q and K are both binary spike tensors, the subtraction $Q_{t,c,i} - K_{t,c,i}$ reduces to an addition and inversion operation on spikes, which can be implemented efficiently on neuromorphic hardware using simple adder circuits.

### C.3 COMPARISON WITH SPIKING SELF-ATTENTION

Spiking self-attention (SSA) was introduced in Spikformer to mimic the dot-product attention mechanism in spiking neural networks. SSA converts query, key and value into spike-form tensors and computes an attention weight matrix via spiking operations followed by a scaling factor. Although SSA removes floating-point multiplications and the softmax, it still forms a full similarity matrix across all tokens, leading to quadratic complexity in the number of points or channels. Moreover, SSA requires three spike-form components (Q, K and V), and its output variance must be scaled to avoid vanishing gradients.

SVMT departs from SSA in two fundamental ways:

- **Local, channel-wise comparison.** Instead of computing a dot product between the query and key across all points, SVMT compares Q and K channel by channel at each point via the difference operator. There is no need to form a large similarity matrix; each entry $A_{t,c,i}$ depends only on the local difference between $Q_{t,c,i}$ and $K_{t,c,i}$. This yields linear complexity $\mathcal{O}(TCN)$ rather than the quadratic $\mathcal{O}(TN^2)$ complexity of SSA.

- **Reduced synaptic operations and energy consumption.** The difference operator $Q-K$ is implemented through addition of binary spike streams, whereas SSA involves spike-based matrix multiplications. The Q–K design therefore uses only two spike-form components (Q and K) and avoids the value component entirely, which markedly reduces synaptic operations. According to the energy analysis of Q-K attention, masking and addition operations can be implemented with negligible power consumption on neuromorphic chips, making difference-based attention substantially more efficient than SSA.

In summary, by using the Q–K difference as a per-channel gate, our SVMT captures salient variations in point cloud features without forming dense attention matrices. This design aligns with the sparse, irregular nature of point clouds and provides significant computational and energy advantages over spiking self-attention.

### C.4 DYNAMIC MASK DEFINITIONS

Two masking strategies are considered. These do not operate simultaneously; instead, the architecture uses only one of them depending on the variant.

#### C.4.1 POINT MASK ($M_{\mathrm{p}}$)

In the *SpikeNet-N* variant, a mask is derived per point by summing the attention activations over time and channels:

$$s_{\mathrm{p}}(i) = \sum_{t=1}^{T} \sum_{c=1}^{C} A_{t,c,i}, \qquad M_{\mathrm{p}}(i) = \begin{cases} 1, & s_{\mathrm{p}}(i) \geq \theta_{\mathrm{p}}, \\ 0, & \text{otherwise}, \end{cases} \qquad (12,13)$$

where $\theta_{\mathrm{p}}$ is a threshold (typically 1). A point $i$ is preserved if it triggers attention in at least one channel and time step. Otherwise it is masked out and its value does not propagate further.

---

**Algorithm 1** Point Mask ($M_{\mathrm{p}}$) Calculation

---

**Require:** Attention tensor $A \in \{0,1\}^{T \times C \times N}$, threshold $\theta_{\mathrm{p}}$
**Ensure:** Point mask $M_{\mathrm{p}} \in \{0,1\}^N$
1: **for** $i = 1$ to $N$ **do**
2:      $s_{\mathrm{p}}(i) = \sum_{t=1}^{T}\sum_{c=1}^{C} A_{t,c,i}$          ▷ Sum activations over time and channels
3:      **if** $s_{\mathrm{p}}(i) \geq \theta_{\mathrm{p}}$ **then**
4:          $M_{\mathrm{p}}(i) = 1$          ▷ Retain the point
5:      **else**
6:          $M_{\mathrm{p}}(i) = 0$          ▷ Mask the point
7:      **end if**
8: **end for return** $M_{\mathrm{p}}$

---

### C.4.2   CHANNEL MASK ($M_{\mathrm{c}}$)

In the *SpikeNet-C* variant, a mask is derived per channel by summing the attention activations over time and points:

$$s_{\mathrm{c}}(c) = \sum_{t=1}^{T}\sum_{i=1}^{N} A_{t,c,i}, \qquad M_{\mathrm{c}}(c) = \begin{cases} 1, & s_{\mathrm{c}}(c) \geq \theta_{\mathrm{c}}, \\ 0, & \text{otherwise}, \end{cases} \tag{14,15}$$

where $\theta_{\mathrm{c}}$ is a threshold. A channel is retained if it has a non–zero attention activation across the dataset; channels with no attention are masked out.

---

**Algorithm 2** Channel Mask ($M_{\mathrm{c}}$) Calculation

---

**Require:** Attention tensor $A \in \{0,1\}^{T \times C \times N}$, threshold $\theta_{\mathrm{c}}$
**Ensure:** Channel mask $M_{\mathrm{c}} \in \{0,1\}^C$
1: **for** $c = 1$ to $C$ **do**
2:      $s_{\mathrm{c}}(c) = \sum_{t=1}^{T}\sum_{i=1}^{N} A_{t,c,i}$          ▷ Sum activations over time and points
3:      **if** $s_{\mathrm{c}}(c) \geq \theta_{\mathrm{c}}$ **then**
4:          $M_{\mathrm{c}}(c) = 1$          ▷ Retain the channel
5:      **else**
6:          $M_{\mathrm{c}}(c) = 0$          ▷ Mask the channel
7:      **end if**
8: **end for return** $M_{\mathrm{c}}$

---

The masked attention output is then

$$E_{t,c,i} = V_{t,c,i}\, A_{t,c,i} \begin{cases} M_{\mathrm{p}}(i), & \text{for SpikeNet-N}, \\ M_{\mathrm{c}}(c), & \text{for SpikeNet-C}. \end{cases} \tag{16}$$

### C.5   ADAPTIVE BEHAVIOR ON SPARSE POINT CLOUDS

Assume that $Q$ and $K$ have spike activation probability $p$, meaning that an entry in $Q$ or $K$ is 1 with probability $p$. The probability that a vector difference triggers a spike in the attention matrix is then approximately $\beta p^2$, where $\beta$ depends on the threshold in $\tau$ and SN.

### C.5.1   POINT MASK VARIANT

For SpikeNet-N, a point is selected if it has at least one attention activation across all channels and time steps. With $\theta_{\mathrm{p}} = 1$ the mask reduces to a logical OR:

$$M_{\mathrm{p}}(i) = 1 - \prod_{t=1}^{T}\prod_{c=1}^{C}(1 - A_{t,c,i}). \tag{17}$$

Hence the probability that point $i$ is kept is

$$\Pr\big(M_{\mathrm{p}}(i) = 1\big) = 1 - (1 - \beta p^2)^{TC} \approx \beta p^2\, T\, C. \tag{18}$$

Therefore, the expected number of retained points decreases linearly with the spike rate $p$. Since computations are performed only on the retained points, the overall complexity adapts to the sparsity of the input.

### C.5.2 CHANNEL MASK VARIANT

In SpikeNet-C, a channel is retained if it fires at least once across all points and time steps. With $\theta_c = 1$, we have

$$M_c(c) = 1 - \prod_{t=1}^{T} \prod_{i=1}^{N} (1 - A_{t,c,i}), \tag{19}$$

and the probability that channel $c$ is retained is

$$\Pr\big(M_c(c) = 1\big) = 1 - (1 - \beta p^2)^{TN} \approx \beta p^2 \, T \, N. \tag{20}$$

Thus the number of active channels drops with the square of the spike rate. In sparse scenarios most channels are pruned, greatly reducing computation without discarding channels that represent salient features.

### C.5.3 EXPECTED COMPLEXITY

Define the effective attention elements as those entries $(t, c, i)$ where $A_{t,c,i} = 1$ and the mask allows propagation. We obtain the following propositions. Suppose $Q$ and $K$ have spike activation rate $p$ and the attention activation rate is $\beta p^2$. Then, for sufficiently small $p$, the expected number of effective attention elements is approximately:

$$\mathbb{E}\big[\text{effective elements}\big]_{\text{SpikeNet-N}} \approx \beta p^2 \, T \, C \, \mathbb{E}\big[\{i : M_p(i) = 1\}\big], \tag{21}$$

$$\mathbb{E}\big[\text{effective elements}\big]_{\text{SpikeNet-C}} \approx \beta p^2 \, T \, N \, \mathbb{E}\big[\{c : M_c(c) = 1\}\big]. \tag{22}$$

Since $\mathbb{E}[\{i : M_p(i) = 1\}] \approx \beta p^2 \, T \, C \, N$ and $\mathbb{E}[\{c : M_c(c) = 1\}] \approx \beta p^2 \, T \, N \, C$ for small $p$, it follows that

$$\mathbb{E}\big[\text{effective elements}\big]_{\text{SpikeNet-N}} \approx \beta^2 p^4 \, T^2 \, C^2 \, N, \tag{23}$$

$$\mathbb{E}\big[\text{effective elements}\big]_{\text{SpikeNet-C}} \approx \beta^2 p^4 \, T^2 \, N^2 \, C. \tag{24}$$

Thus the complexity declines roughly with $p^4$ in highly sparse regimes. This behaviour explains why SpikeNet-N and SpikeNet-C achieve high energy efficiency on sparse point clouds without significantly compromising important information.

### C.6 SUMMARY OF SVMT

The proposed SVMT introduces a binary attention mechanism within spiking neural networks and equips it with either a point mask (SpikeNet-N) or a channel mask (SpikeNet-C) to adaptively process sparse point clouds. By masking inactive points or channels, the network reduces its computational burden in proportion to the sparsity of the input, as demonstrated by the above mathematical analysis. These theoretical results support the empirical findings reported in the main SpikeNet paper and clarify how the dynamic masks contribute to the energy efficiency of the architecture.

## D DATASET INTRODUCTION

ModelNet40 Wu et al. (2015) is an important benchmark dataset in the field of point cloud processing. It includes 40 common 3D object categories, such as airplanes, chairs, etc., and the data comes from CAD models and 3D scanning data. The dataset consists of approximately 12,000 3D models, with 9,800 used for training and 2,200 for testing. This dataset is simple and standardized, and is widely used for training and evaluating 3D object classification algorithms.

ScanObjectNN Uy et al. (2019) is an object classification dataset based on real-world 3D scanning data. The data comes from 3D scans of various real-world scenes, including both indoor and outdoor objects. The dataset contains 280,414 point cloud samples, divided into 15 object categories. One of the main features of this dataset is the diversity in sampling density, noise levels, and missing data, which highly simulates the complexity of real-world 3D scanning data. It is mainly used to evaluate the performance of point cloud classification algorithms under practical complex data, helping to verify the reliability of algorithms in real-world applications.

ShapeNetPart Yi et al. (2016) is a subset of the ShapeNet dataset, focusing on the task of 3D object part segmentation. It is based on the ShapeNetCore dataset, with part annotations made on its models. It covers 16 object categories and contains 16,881 3D models. In addition to geometric data, the dataset also includes detailed part annotation information in the .json format. ShapeNetPart plays an important role in research on 3D object part segmentation algorithms, providing strong support for training neural networks for accurate part segmentation of objects.

S3DIS Armeni et al. (2016) is an important benchmark dataset in the field of 3D indoor scene semantic segmentation. It is built based on lidar scanning data of multiple indoor spaces, covering 6 typical indoor scene types such as offices, meeting rooms, corridors, and laboratories. The dataset contains point cloud data of 12 large-scale indoor areas, which are subdivided into 271 room-level sub-regions, with a total number of point clouds exceeding 2 billion. Each point in the dataset is annotated with detailed semantic categories, involving 13 common indoor object and structure categories such as walls, floors, ceilings, furniture, doors, and windows, providing precise label support for semantic segmentation tasks.

Semantic KITTI Behley et al. (2019) is an outdoor 3D semantic segmentation benchmark derived from KITTI. Recorded by a 64-layer vehicle-mounted LiDAR, it compiles 22 temporally-ordered sequences spanning urban, rural and highway scenes. More than 43 000 full-spin scans are annotated with 28 classes—vehicles, pedestrians, cyclists, roads, buildings, etc.—preserving scene dynamics and spatio-temporal coherence. The dataset is now the standard for training and evaluating LiDAR-based segmentation and detection algorithms in autonomous driving.

ScanNet V2 Dai et al. (2017) is an indoor RGB-D 3D semantic segmentation dataset for 3D scene understanding. Its data is captured by RGB-D cameras (recording visual appearance and depth information) across 1,513 annotated indoor scenes, defining 20 semantic categories covering common indoor objects like furniture and structural elements. Featuring high-quality 3D reconstruction, dense semantic annotations, and multimodal data, it is widely used as a benchmark for training and evaluating indoor 3D semantic segmentation, object detection, and scene understanding algorithms.

# E  EXPERIMENT DETAILS

This section will provide a detailed introduction to the specific experimental setup and some supplementary experimental results.

## E.1  ANALYSIS ON DIFFERENT ACTIVATION FUNCTIONS

During the direct training of SpikeNet, the gradient of the spiking activation function used during backpropagation is based on the arctangent function (ATan):

$$g(x) = \frac{1}{\pi} \arctan\left(\frac{\pi}{2}\alpha x\right) + \frac{1}{2} \tag{25}$$

Hence, the gradient for backpropagation is:

$$g'(x) = \frac{\alpha}{2\left(1 + \left(\frac{\pi}{2}\alpha x\right)^2\right)} \tag{26}$$

In the direct training of SNNs, the commonly used spiking activation function is the Sigmoid. To verify the network performance, we conduct ablation experiments, keeping all other network configurations unchanged. The results are shown in the table 12. Our SpikeNet with the arctangent function demonstrating the best performance.

## E.2  ANALYSIS ON DIFFERENT SPIKING NEURONS

The core of SNN lies in spiking neurons. Models utilizing Integrate-and-Fire (IF)Bulsara et al. (1996) neurons—which only require accumulation and threshold comparison operations—are not only easy to deploy on neuromorphic chips but also feature extremely low energy consumption, making them particularly suitable for resource-constrained edge devices. In contrast, the leakage

Table 12: Performances of different activation functions on the ModelNet40 dataset.

| Function | OA (%) | mAcc (%) |
|---|---|---|
| Sigmoid | 93.2 | 90.6 |
| ATan | **93.4** | **90.9** |

Table 13: Ablation on Different Spiking Neurons on the ModelNet40 dataset.

| Spiking Neurons | OA (%) | mAcc (%) |
|---|---|---|
| LIF | 92.6 | 90.1 |
| IF | **93.3** | **90.9** |

mechanism of Leaky Integrate and-Fire (LIF)Gerstner & Kistler (2002) neurons necessitates additional circuits to simulate the decay process of membrane potential, which increases hardware complexity and energy consumption. We conducted ablation experiments on different spiking neurons, and the results, as shown in Table 13, indicate that the SpikeNet-C model combining IF neurons with the spiking attention mechanism achieves optimal performance.

### E.3 ANALYSIS ON SPIKE THRESHOLD

For SNN, the threshold $v_{\text{threshold}}$ is a critical parameter. We conducted ablation experiments on this threshold (with results shown in Table 14), testing four values: 0.5, 1.0, 1.5, and 2.0. When training SpikeNet on the ModelNet40 dataset, the model performance shows a trend of first improving and then declining as the threshold changes. A too-low threshold (e.g., 0.5) may cause excessive activation of neurons, introducing redundant noise signals that interfere with the extraction of effective features; while a too-high threshold (e.g., 1.5, 2.0) will suppress neuron activation, leading to the loss of some key feature information and thus limiting classification performance. In contrast, the threshold of 1.0 can balance activation intensity and feature retention capability, enabling neurons to effectively respond to key features while avoiding noise interference caused by excessive activation, achieving the optimal performance (OA of 93.4% and mAcc of 90.9%).

Table 14: Performances of different thresholds on the ModelNet40 dataset.

| Threshold | OA (%) | mAcc (%) |
|---|---|---|
| $v_{\text{threshold}} = 0.5$ | 93.0 | 90.5 |
| $v_{\text{threshold}} = 1.0$ | **93.4** | **90.9** |
| $v_{\text{threshold}} = 1.5$ | 92.8 | 90.4 |
| $v_{\text{threshold}} = 2.0$ | 92.6 | 90.1 |

### E.4 DATASET CONFIGURATION AND DATA AUGMENTATION

For the ModelNet40 dataset, during the training process, we uniformly sample each point cloud object to 1024 points. We enhance the data through random translation in the range of [-0.2, 0.2], random anisotropic scaling in the range of [0.67, 1.5], and random input dropout. In the experiments on the ScanObjectNN dataset, we selected the most challenging perturbation variable (PB_T50_RS) and applied only scaling for data augmentation. During training, we used a batch size of 32 for both datasets, with a total of 400 epochs, and adjusted the learning rate for each epoch using a cosine annealing scheduler.

Table 15: Performance comparisons of different data augmentation methods on both the ModelNet40 and the ScanObjectNN datasets.

| Methods | ModelNet40 | | ScanObjectNN | |
|---|---|---|---|---|
| | OA (%) | mAcc (%) | OA (%) | mAcc (%) |
| Random-TSD | **93.4** | **90.9** | 84.0 | 81.6 |
| Random-S | 92.6 | 89.6 | **85.5** | **83.3** |

We define two different data augmentation methods: random translation-scaling-dropout (Random-TSD) and random scaling (Random-S). We also conduct comparative experiments on different datasets, with the results shown in the table 15. On ModelNet40, our SpikeNet-C with the Random-TSD achieves the improved performance, while SpikeNet-C with the Random-S performs better on ScanobjectNN.

### E.5 THE CHOICE OF K IN THE GAM MODULE

In the GAM module, the neighbor-number parameter k is crucial for the sampling and extraction of local-point features, as it determines the size of the perception area of local points. We report the experimental results with different values of k in Table 16. The results show that when k=24, the point cloud classification accuracy is the highest.

Table 16: Effect of using a different number of neighbors for local-based edge point sampling.

| $k$ | 16 | 24 | 32 | 48 |
|---|---|---|---|---|
| OA (%) | 92.9 | **93.4** | 93.1 | 92.8 |

Table 17: Effect of using different sizes points sampling.

| $N$ | 512 | 1024 | 2048 | 4096 |
|---|---|---|---|---|
| OA (%) | 92.7 | **93.4** | 93.0 | 92.9 |

### E.6 THE SAMPLING POINTS OF DIFFERENT SIZES

At present, most deep-learning-based 3D point cloud networks uniformly sample each point cloud to 1024 points for experiments. To explore the impact of sampling points of different scales on the performance of 3D point cloud classification networks, we conducted ablation experiments and reported the results with 2048 and 4096 points in Table 17.

### E.7 ANALYSIS ON DSSR MODULE

To highlight the innovation of the proposed Dynamic Sparse Spiking Residual (DSSR) structure in adapting to 3D point clouds, we conducted ablation experiments by replacing DSSR with typical 2D spiking residual modules. These 2D residual structures adopt a single-stage static design, with their core formula expressed as:

$$Y^{(l)} = \text{BN}(\text{Conv}(\text{SN}(\text{BN}(\text{Conv}(\text{SN}(X^{(l)}))))))+ X^{(l)}. \tag{27}$$

Here, the input feature $X$ is directly converted into spike sequences through a single Spiking Neuron (SN) layer, and the residual connection is established between the original input X and the output of the two-layer Conv-SN stack. In the experiment, we retained all other components of SpikeNet (e.g., SVMT, GAM, and AMP) and only replaced DSSR with the 2D spiking residual structure. The results on the ModelNet40 dataset are shown in Table 18. DSSR outperforms the 2D spiking residual structure by 1.2% in OA and 1.6% in mAcc. This is because the single SN layer in 2D residual structures cannot dynamically suppress invalid spikes in sparse regions, leading to redundant computations, and thus fail to adapt to 3D point cloud structures. In contrast, DSSR's design balances the preservation of point cloud features and spiking sparsity, making it more suitable for 3D point clouds.

Table 18: Analysis on DSSR Module on ModelNet40.

| Residual Structure | OA (%) | mAcc (%) |
|---|---|---|
| Spiking Residual | 92.2 | 89.3 |
| DSSR | **93.4** | **90.9** |

### E.8 ANALYSIS ON SVMT MODULE

To verify the superiority of the proposed Spiking Vector Mask Transformer (SVMT) in adapting to point cloud features, we conducted ablation experiments on the ModelNet40 dataset by replacing SVMT with attention modules from SNN works oriented to 2D images: the token attention

from QKformer Zhou et al. (2024a) and the spike self-attention (SSA) from Spikformer Zhou et al. (2023b). All other components (e.g., DSSR, GAM, and AMP) remained unchanged to ensure a fair comparison. The experimental results are shown in Table 19. The SVA module outperforms QKformer's token attention and Spikformer's SSA by 1.5% and 1.3% in OA, respectively. This discrepancy stems from the inherent differences between 3D point clouds and 2D images: 2D images consist of dense grid-structured pixels with strong spatial correlation, thus making QKformer's scalar attention and Spikformer's SSA effective. However, point clouds are characterized by sparsity, disorder, and lack of fixed topology. SVMT, by dynamically aligning with the sparsity of point clouds through binary spiking masks and capturing geometric differences between 3D points using channel subtraction-based vector attention, can focus on meaningful features and suppress noise, thereby achieving higher accuracy.

Table 19: Analysis on SVMT Module on ModelNet40.

| Attention Module | OA (%) | mAcc (%) |
| --- | --- | --- |
| Token Attention | 91.9 | 88.6 |
| Spike Self-Attention | 92.1 | 89.4 |
| Spiking Vector Attention | **93.4** | **90.9** |

### E.9 ANALYSIS ON STAGE NUMS

To further verify the effectiveness of the proposed modules under parameter-constrained settings, we conducted experiments on the ModelNet40 dataset by adjusting the number of stages in the Spiking Encoder of SpikeNet-C to control the total parameters. The experimental results are shown in the table 20. When SpikeNet-C only performs encoding with one stage, the model parameter count is only 0.3M, and the accuracy still exceeds E-3DSNN'sQiu et al. (2025) 91.7%, reaching 91.9%. This indicates that even with extremely low parameter consumption, our model is still superior to existing state-of-the-art snn methods, validating that the proposed DSSR and SVMT modules can effectively enhance feature representation capability without relying on excessive parameter overhead. When SpikeNet-C performs encoding with four stages, the performance improves to 93.4%, which demonstrates that the hierarchical feature enhancement mechanism of our architecture can effectively utilize additional parameters to further improve accuracy. This incremental improvement further confirms that the performance gains of SpikeNet stem from the rational design of its core modules.

Table 20: Analysis of Stage Nums on ModelNet40.

| Stage Nums | Parameter (M) | OA (%) | mAcc (%) |
| --- | --- | --- | --- |
| 1 | 0.3 | 91.9 | 88.4 |
| 2 | 0.9 | 92.4 | 89.6 |
| 3 | 2.9 | 92.8 | 90.1 |
| 4 | 10.5 | **93.4** | **90.9** |

### E.10 ANALYSIS ON SPIKENET VARIANTS

We further investigated two variants of SpikeNet, namely SpikeNet-N and SpikeNet-C: SpikeNet-N is capable of masking irrelevant and redundant points in point clouds, while SpikeNet-C can filter out redundant channel information. To verify the effect of superimposing these two variants, we kept other experimental parameters unchanged and superimposed them in different orders to define two structures: SpikeNet-NC (N first, then C) and SpikeNet-CN (C first, then N). Ablation experiments were conducted based on these two superimposed structures. The experimental results, as shown in Table 21, indicate that after multi-layer spike propagation, the feature information of both SpikeNet-NC and SpikeNet-CN becomes extremely sparse, ultimately leading to a significant drop in model accuracy.

Table 21: Analysis of SpikeNet Variants on ModelNet40.

| Model | OA (%) | mAcc (%) |
|---|---|---|
| SpikeNet-NC | 91.6 | 88.2 |
| SpikeNet-CN | 92.0 | 88.7 |
| SpikeNet-N | 93.2 | 90.4 |
| SpikeNet-C | **93.4** | **90.9** |

### E.11 THE DETAILED DESIGN OF SPIKE DECODER

We adopted the PointNet++ architecture to perform part segmentation. Regarding the design of each module in the Spike Decoder, we reported the experimental results in Table 22. Compared with the structure based on Artificial Neural Network (ANN), the performance of our SpikeNet has a difference of less than 1%. Moreover, the design using Spiking Neural Networks has the advantage of energy-saving.

Table 22: Performances of different Spike Decoder designs on the ShapeNetPart dataset,MLP* and Upsample* represent Conv-BN-ReLU based on ANN.

| Spike Decoder | Cat.mIoU | Ins.mIoU |
|---|---|---|
| Upsample-SVMT | 83.3 | 85.1 |
| Upsample-DSSR | 83.9 | 85.4 |
| Upsample*-MLP* | 84.2 | 85.6 |

### E.12 FIRING RATES OF DIFFERENT NETWORK LAYERS

We present the firing rates of Query, Key, and Value for both the SVA module and the SSA module at different stages of the Spike Encoder in a single figure. As shown in Figure F, the Query, Key, and Value of both modules exhibit extremely high sparsity, which provides a foundation for the sparse computation of the SVMT (presumably an SVA-based module, such as Spike Vector Matching Transformer) and the SpikeNet with the SSA module. This is also the core of the energy-saving advantage of spiking neural networks. Furthermore, it can be clearly observed from the figure that under the training scenario on the ModelNet40 dataset, the proposed SVMT design (corresponding to the SVA module) has an overall lower firing rate compared to the SpikeNet with the SSA module. This characteristic makes it more suitable for the processing task of unordered point clouds.

## F ENERGY CONSUMPTION

The proposed network is designed with biological plausibility in mind. Ideally, it can be directly deployed on neuromorphic hardware. The inference energy cost of our SpikeNet can be expressed as:

$$E_{\text{SpikeNet}} = E_{\text{MAC}} \times (\text{FLOPs}_{\text{Conv}}^{\text{E}} + \text{FLOPs}_{\text{FC}}) + E_{\text{AC}} \cdot T \sum_{n=1}^{N} \text{FLOPs}_{\text{SConv}}^{\text{n}} \cdot \text{fr}^{\text{n}}. \tag{28}$$

Where $\text{FLOPs}_{\text{Conv}}^{\text{E}}$ represents the floating-point operations of the floating-point convolutional layer in the Embedding module, and $\text{FLOPs}_{\text{FC}}$ represents the FLOPs of the fully connected (FC) layer. $\text{FLOPs}_{\text{SConv}}^{\text{n}}$ represents the FLOPs of the spiking convolutional layer, where $N$ is the total number of spiking convolutional layers, $T$ is the time step, and $\text{fr}^{\text{n}}$ is the firing rate of the $N$-th spiking convolution. We assume that the MAC and AC operations are implemented on SpikingJellyFang et al. (2023), where $E_{\text{MAC}} = 4.6\,\text{pJ}$ and $E_{\text{AC}} = 0.9\,\text{pJ}$(Kundu et al., 2021b; Hu et al., 2021; Horowitz, 2014; Kundu et al., 2021a).

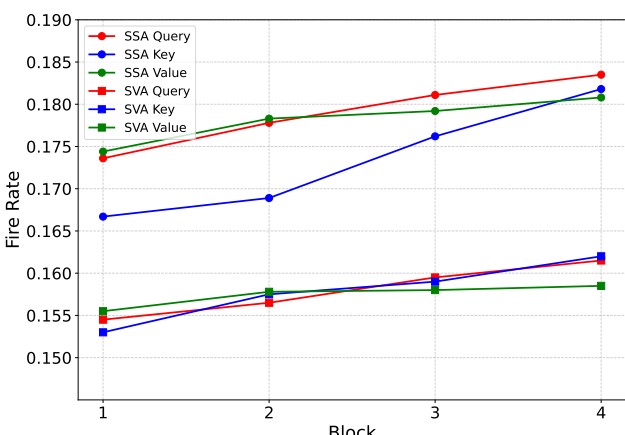

Figure 7: The firing rates of Q, K, V in the SVA module and SSA module.

## G    THE DETAILED SPIKING COMPUTATION MECHANISM OF SPIKENET

To gain a deeper understanding of the spiking computation mechanism of the SpikeNet network model, we will explain in detail the specific network layers in each stage of the Spike Encoder of the classification network. The GAM module, similar to the sampling and grouping operation in PointNet++, aims to extract lightweight local point features without involving any MAC operations. When the DSSR module receives the local point features, it first uses the Spiking Neuron layer (SN) to convert them into spike signals and then performs the Conv operation. This process fully complies with the rules of spiking computation. Although the output after passing through the two-layer residual structure is a floating-point number, the subsequent AMP operation only conducts pooling and does not include MAC operations. Subsequently, when the previous floating-point values go to the SVMT module, the SN layer first converts them into spike sequences, and then the Conv operation takes place. Obviously, before performing the Conv operation involving MAC in each stage, we convert the floating-point values into spikes, strictly following the spiking principle, which greatly reduces the energy consumption.

For the Decoder of the segmentation network, we also use the Upsample module composed of SN-Conv-BN connected in sequence. First, the SN layer converts the floating-point numbers into spike signals, and then we perform the Conv operation. The DSSR module functions similarly. It can be seen that our entire network strictly adheres to the principle of spiking computation.

## H    NEUROMORPHIC CHIP DEPLOYMENT

We expect to directly deploy SpikeNet on neuromorphic hardware. However, designing RC circuits on neuromorphic hardware for processing 3D point cloud requires additional engineering efforts. Point cloud data typically exists in floating-point format. It is a natural choice to use the Embedding method with 'multiply-accumulate (MAC)' operations to generate spike signals for each point. Moreover, to pursue higher classification accuracy, the final fully connected (FC) layer needs to perform some MAC operations, which are quite difficult to implement on neuromorphic hardware. Currently, there is no method that combines laser scanning with an event-driven mechanism to generate 3D point cloud spikes from the source. Theoretically, however, both our Spike Encoder and Spike Decoder can be implemented on neuromorphic hardware. They can convert the floating-point point clouds into spike form before performing convolution (Conv) operations and execute accumulation (AC) operations to accumulate the weights of postsynaptic neurons. The 3D-SNN architecture can transform matrix multiplication into sparse addition, enabling addressable addition on neuromorphic chips.

## I    LLM USAGE STATEMENT

In accordance with the policy requirements of the ICLR 2026 Conference regarding the use of Large Language Models (LLMs), the authors of this paper hereby declare that: All core academic content of this research—including the formulation of research ideas, design of methodologies, collection and analysis of experimental data, derivation of results, and refinement of conclusions—was independently completed by human authors in its entirety, with no involvement of any LLM tools.

During the paper writing process, the authors did not use LLMs for content generation, code writing, literature analysis, or result verification. Grammar checking of text expression and format adjustment were all completed manually, without assistance from any LLM tools such as ChatGPT, GPT-4, or Claude.

The authors confirm that there is no undisclosed LLM usage in this research, and all content complies with the requirements for academic integrity specified in the ICLR Code of Ethics. The authors bear full responsibility for the authenticity, accuracy, and originality of this paper, and have verified through manual checks that there are no factual errors or plagiarism risks.

## J    ETHICS STATEMENT

We confirm that all authors have read and comply with the ICLR Code of Ethics. Our work does not involve human subjects, sensitive data collection or release, potentially harmful methodologies or applications, conflicts of interest, discrimination/bias/fairness concerns, privacy/security issues, legal compliance challenges, or research integrity matters that require explicit addressing.

## K    REPRODUCIBILITY STATEMENT

To ensure the reproducibility of our results, we will make our implementation code available as supplementary material. The datasets used (e.g., ModelNet40, ScanObjectNN, ShapeNetPart, S3DIS for point cloud tasks) are publicly accessible, with relevant details provided in the main text. Key experimental settings (including hyperparameters like SNN threshold and timestep optimized in our analysis) and procedures are thoroughly documented in the manuscript. Any theoretical derivations supporting our work are included in the appendix.

