# OpenReview forum: "SpikeNet: Sparse Spike-Driven Mask Vector Transformer for Energy-Efficient and Stable Spiking Point Cloud Processing"
_ICLR.cc/2026/Conference — Submitted to ICLR 2026_

### Official Review · Reviewer_j5DG · 2025-10-26

**Soundness:** 2
**Presentation:** 3
**Contribution:** 1
**Rating:** 4
**Confidence:** 4

**Summary:**

This paper identifies the trade-off between accuracy and efficiency in existing Spiking Neural Networks (SNNs) for point cloud processing. To address this, the authors propose SpikeNet, a novel SNN architecture featuring a Spiking Vector Mask Transformer (SVMT) that uses binary spike masks to eliminate costly Softmax and multiplication operations, and a Dynamic Sparse Spiking Residual (DSSR) structure to stabilize training. Experiments show that SpikeNet achieves state-of-the-art performance among SNNs on classification and segmentation tasks, competitive with conventional Artificial Neural Networks (ANNs).

**Strengths:**

1. The proposed SVMT module is well-justified, as it cleverly leverages binary spiking masks to dynamically align with the native sparsity of point cloud data, providing a natural and efficient alternative to standard attention.

2. This design provides a significant energy-saving advantage by fundamentally eliminating the need for the computationally intensive softmax and multiplication operations that are typical in transformer-based models.

**Weaknesses:**

The main weakness of this paper is its unclear application value.

1. The authors mainly compare the energy consumption in their experiments. However, in most real-world application, inference latency and memory consumption are more important. The authors should provide these results.

2. How does the energy calculated? Any hardware experiments to validate its impact?

3. The authors only conduct experiments on shape-level 3D analysis, which is too toy to validate the real-world application value. Shape analysis is very different from scene-level perception, so at least experiments on scene segmentation should be provided.

**Questions:**

See weakness. Shape-level pointcloud analysis is a very toy experimental setting. For pointcloud analysis, scene-level perception is much closer to real application.

---

> ### Author Response · Authors · 2025-11-23
> **Rebuttal-1**
>
> ***Q1. The authors mainly compare the energy consumption in their experiments. However, in most real-world application, inference latency and memory consumption are more important. The authors should provide these results.***
>
> **A1.** Thank you for this valuable suggestion. We completely agree that inference latency and memory consumption are critical metrics for real-world deployment. We have added a new experiment in the revised manuscript (**see Table 10**). We compared training and inference overhead on ModelNet40, focusing on two metrics: Latency and Peak Memory. As shown in Table 10, when the number of sampled points increases from 512 to 1024, the latency and memory usage of SNN-based SPT series models rise significantly. Mamba3D, which is based on ANN, also maintains a high resource consumption level. In contrast, our proposed SpikeNet reduces training latency to 321 ms and memory usage to 11.3 GB; in the inference phase, it only requires 185 ms and 6.1 GB. These results are better than existing comparison methods, verifying that SpikeNet balances time efficiency and hardware friendliness while maintaining accuracy.
>
> **Table 10**
> | Methods    | Training           | | Inference          | |
> |------------|--------------------|--------------------|--------------------|--------------------|
> |            | Latency | Memory   | Latency | Memory   |
> | SPT-512    | 326 ms  | 9.7 GB   | 191 ms  | 5.2 GB   |
> | SPT-768    | 385 ms  | 12.5 GB  | 201 ms  | 7.3 GB   |
> | SPT-1024   | 431 ms  | 15.2 GB  | 227 ms  | 9.5 GB   |
> | Mamba3D    | 433 ms  | 14.9 GB  | 256 ms  | 7.8 GB   |
> | SpikeNet(our)| **321 ms** | **11.3 GB** | **185 ms** | **6.1 GB** |
>
> ***Q2. How does the energy calculated? Any hardware experiments to validate its impact?***
>
> **A2.** Thank you for this insightful question regarding energy calculation.
> The operational parameters in the current estimation model (e.g., MAC operation energy consumption of 4.6 pJ, AC operation energy consumption of 0.9 pJ) refer to the measured and calibrated values for the Loihi chip from published works (**see Kundu et al., 2021a [1] and Horowitz, 2014[2]**). We trained the model on the A6000 GPU, ensuring that the estimation results have a quantifiable correspondence with the actual hardware energy consumption. These values follow the operational energy consumption characteristics of real hardware, ensuring that the estimation results have practical reference significance. Detailed introductions to neuromorphic chip energy consumption calculation are available in **Spikformer[3]**. Additionally, **Appendix G** provides a detailed explanation of the model's details, which is in line with biological plausibility. **Appendix H** also describes the reliability of neuromorphic hardware and the challenges in point cloud applications.
>
> [1]Kundu, S., Datta, G., Pedram, M.,  Beerel, P. A. (2021). Spike-thrift: Towards energy-efficient deep spiking neural networks by limiting spiking activity via attention-guided compression. Proceedings of the IEEE/CVF Winter Conference on Applications of Computer Vision (WACV)
>
> [2]Horowitz, M. (2014). 1.1 computing's energy problem (and what we can do about it). 2014 IEEE International Solid-State Circuits Conference Digest of Technical Papers (ISSCC), 10–14.
>
> [3]Spikformer: When Spiking Neural Network Meets Transformer. International Conference on Learning Representations (ICLR).

---

> ### Author Response · Authors · 2025-11-23
> **Rebuttal-2**
>
> ***Q3. The authors only conduct experiments on shape-level 3D analysis, which is too toy to validate the real-world application value. Shape analysis is very different from scene-level perception, so at least experiments on scene segmentation should be provided.***
>
> **A3.** Thank you for this critical comment regarding the need for scene-level validation! We fully agree that real-world application value requires evaluation beyond object-level classification. Therefore, we have conducted a comprehensive experiments on S3DIS (indoor large-scale scene), SemanticKITTI (outdoor self-driving scene) and Scannetv2 (indoor large-scale scene) in the revised main text, where the results on different datasets are shown in Table 3, Table 4 and Table 5, respectively.
>
> **(1) S3DIS:** We have moved the S3DIS semantic segmentation results from **Appendix E.12** to the **main text (Table 3)**, and updated the relevant results of the PT series and the energy consumption estimation of related methods.
>
> **(2) SemanticKITTI:** We compared SpikeNet with other SNN-based and ANN-based methods, using mIoU (validation/test set) as the metric. Among SNN methods, our SpikeNet-N (64.6\%/69.8\%) and SpikeNet-C (65.1\%/70.2\%) outperform E-3DSNN (63.2\%/69.4\%). Due to the mature development, refined optimization of ANNs and no constraints from spiking encoding, the mIoU of SpikeNet is slightly lower than that of advanced ANN methods such as PTv3 (75.5\% on test set).
>
> **(3) Scannet V2** We compared SpikeNet with representative ANN-based and SNN-based methods. The evaluation metric is mean Intersection over Union (mIoU, validation set/test set). Among SNN methods, our proposed SpikeNet-N (69.4\%/70.8\%) and SpikeNet-C (69.7\%/71.2\%) perform better than E-3DSNN (68.2\%/69.5\% on test set), and are only slightly lower than PTv3 (77.5\%/77.9\%), an advanced ANN method.
>
> **Table 3**
> | Method | Type | Type | mIoU | ceiling | floor | wall | beam | column | window | door | table | chair | sofa | bookcase | board | clutter | Energy(mJ) |
> |-|-|-|-|-|-|-|-|-|-|--|-|-|--|-|-|-|-|
> | PointNet|17'CVPR | ANN | 41.1 | 88.8 | 97.3 | 69.8 | 0.0 | 3.9 | 46.3 | 10.8 | 59.0 | 52.6 | 5.9 | 40.3 | 26.4 | 33.2 | 5.5 |
> | PointNet++ |17'NIPS | ANN | 53.5 | 89.4 | 97.7 | 75.4 | 0.0 | 1.8 | 58.3 | 19.5 | 79.0 | 69.2 | 59.1 | 46.2 | 58.7 | 41.6 | 5.5 |
> | PointCNN |18'NIPS | ANN | 57.3 | 92.3 | 98.2 | 79.4 | 0.0 | 17.6 | 22.8 | 62.1 | 74.4 | 80.6 | 31.7 | 66.7 | 62.1 | 56.7 | 324.5 |
> | PointNeXt|22'NIPS | ANN | 70.5 | 94.2 | 98.5 | 84.4 | 0.0 | 37.7 | 59.3 | 74.0 | 83.1 | 91.6 | 77.4 | 77.2 | 78.8 | 60.6 | - |
> | PCM|25'AAAI | ANN | 63.4 | 93.3 | 96.7 | 80.6 | 0.0 | 35.9 | 57.7 | 60.0 | 74.0 | 87.6 | 50.1 | 69.4 | 63.5 | 55.9 | - |
> | PointRWKV | 25'AAAI | ANN | 70.5 | 94.2 | 98.3 | 86.5 | 0.0 | 38.6 | 64.5 | 76.2 | 88.2 | 89.3 | 65.2 | 75.6 | 78.2 | 61.3 | - |
> | PTv1|21'ICCV |ANN | 70.4 | 94.0 | 98.5 | 86.3 | 0.0 | 38.0 | 63.4 | 74.3 | 89.1 | 82.4 | 74.3 | 80.2 | 76.0 | 59.3 | 76.8 |
> | PTv2|22'NIPS |ANN | 71.6 | 93.0 | 98.1 | 86.7 | 0.0 | 48.0 | 62.4 | 76.1 | 88.3 | 87.6 | 77.1 | 79.2 | 77.5 | 59.8 | 400.1 |
> | PTv3|24'CVPR |ANN | 73.6 | 92.4 | 98.3 | 86.6 | 0.0 | 55.8 | 63.7 | 77.1 | 83.8 | 93.3 | 79.1 | 79.4 | 85.4 | 61.7 | 687.7 |
> | E-3DSNN | 25'AAAI | SNN | 67.4 | 95.3 | 98.5 | 82.3 | 0.0 | 28.0 | 55.8 | 71.5 | 81.2 | 89.8 | 69.2 | 76.4 | 67.0 | 61.6 | 14.4 |
> | SpikeNet-N | SNN |— | **68.4** | 89.9 | 91.5 | 83.0 | 0.0 | 47.0 | 60.1 | 74.6 | 79.7 | 86.0 | 69.6 | 71.5 | 80.2 | 57.0 | **10.2** |
> | SpikeNet-C | SNN |— | **68.9** | 90.1 | 92.2 | 83.5 | 0.0 | 47.6 | 60.5 | 75.2 | 80.1 | 86.3 | 70.3 | 72.4 | 80.6 | 57.3 | **10.2** |
>
> **Table 4**
> | Method| Type | Input | Val  | Test |
> |-|-|-|-|-|
> | SPVNAS| ANN  | point | 64.7 | 66.4 |
> | Cylinder3D| ANN  | point | 64.3 | 67.8 |
> | AF2S3Net| ANN  | point | 74.2 | 70.8 |
> | PTv2| ANN  | point | 70.3 | 72.6 |
> | PTv3| ANN  | point | 72.3 | 75.5 |
> | E-3DSNN| SNN  | voxel | 63.2 | 69.4 |
> | SpikeNet-N (our) | SNN | point | 64.6 | 69.8 |
> | SpikeNet-C (our) | SNN | point | 65.1 | 70.2 |
>
> **Table 5**
> | Method| Type | Input | Val  | Test |
> |-|-|--|-|-|
> | PointNeXt| ANN  | point | 71.5 | 71.2 |
> | PointMetaBase| ANN  | point | 72.8 | 71.4 |
> | MinkUNet | ANN  | voxel | 72.2 | 73.6 |
> | PTv2| ANN  | point | 75.4 | 75.2 |
> | PTv3| ANN  | point | 77.5 | 77.9 |
> | E-3DSNN| SNN  | voxel | 68.2 | 69.5 |
> | SpikeNet-N (our) | SNN | point | 69.4 | 70.8 |
> | SpikeNet-C (our) | SNN | point | 69.7 | 71.2 |
>
> ***Q4. Shape-level point cloud analysis is a very toy experimental setting. For point cloud analysis, scene-level perception is much closer to real application.***
>
> **A4.** Thank you very much for your suggestion. We have significantly expanded our evaluation to include scene-level perception (e.g., semantic segmentation). As comprehensively outlined in our response to **Q3**, the revised paper now includes results on S3DIS, SemanticKITTI and ScanNet V2, which firmly establish the practical relevance of our work beyond initial shape-level analysis.

---

> > ### Comment · Reviewer_j5DG · 2025-11-27
> >
> > Thanks for the authors' rebuttal. It solves my major concerns. I will raise my score if other reviewers do not have further questions.

---

### Official Review · Reviewer_NzVp · 2025-10-30

**Soundness:** 3
**Presentation:** 3
**Contribution:** 3
**Rating:** 4
**Confidence:** 4

**Summary:**

The paper proposes SpikeNet, a spiking neural backbone for point-cloud classification and part segmentation that couples a Dynamic Sparse Spiking Residual (DSSR) module with a Spiking Vector Mask Transformer (SVMT). DSSR is designed to stabilize direct SNN training via bounded surrogate gradients and dynamic sparsity, leading to a provable attenuation of gradient amplification. SVMT replaces dot-product attention with a local per-channel Q–K difference gate and a spike-driven mask, which removes softmax and quadratic token interactions; the resulting attention scales linearly in time–channel–point dimensions and is amenable to neuromorphic implementations.

**Strengths:**

1. The proposed Spiking Vector Mask Transformer (SVMT) is not a trivial substitution of ReLU with spikes. It redefines attention itself in spike terms: attention weights are derived from spike-domain channelwise differences $\tau(Q,K)=Q-K$ instead of dense dot products and softmax, and sparsification is enforced through a Spike Point Masker that gates either channels or points using binary spike statistics.
2. The model is benchmarked on ModelNet40, ScanObjectNN, and ShapeNetPart, which together cover CAD-like clean geometry, noisy real-world scans, and fine-grained part-level segmentation.
3. The ablations are not superficial. They analyze alternative attention operators (sum, product, etc.), masking strategies (channel vs point), and embedding dimensionality. These studies make a credible case that the final architecture is not arbitrary.

**Weaknesses:**

1. DSSR’s stability argument is persuasive but qualitative. The paper does not report gradient-norm distributions across depth, training-loss curves comparing with/without DSSR, or any theorem giving a global bound across layers and timesteps. Adding a small empirical study here would turn into strong evidence.
2. The Spike Point Masker is described in both pointwise and channelwise forms, but notation occasionally glosses over which axes are being summed (over points $N$, channels $C$, timesteps $T$). This matters because those masks drive sparsity and thus energy savings.
3. The model has not yet been demonstrated on full outdoor LiDAR scenes, nor is there an ablation of memory scaling in that regime. S3DIS shows meaningful indoor semantics, but the overall mIoU is still behind the best ANN scene-segmentation models.
4. A few phrases imply that SpikeNet “removes multiplications” or is “entirely neuromorphic.” The appendices clarify the nuance: SVMT avoids $QK^\top$ dot-product attention and softmax, but linear projections and masked elementwise products still exist; the encoder/decoder are spike-friendly, but the input embedding and final FC head are still float MAC. These nuances should be stated up front.

**Questions:**

1. Can you provide per-sample inference latency and measured runtime energy or power traces on any available hardware?
2. Can you share gradient-norm vs depth, or loss-vs-epoch curves, comparing SpikeNet with and without DSSR? This would convert the qualitative stability argument into quantitative evidence.
3. Spike Point Masker details: Please precisely state tensor layout ($T\times C\times N$ or equivalent) and specify along which axes you sum to get channel masks vs point masks. A short pseudocode snippet in the appendix would remove any ambiguity.
4. You state that the encoder/decoder can in principle be mapped to spike-friendly AC-heavy hardware, while the input embedding stem and FC head remain MAC-dominated. Is the intended deployment strategy a hybrid (MAC front/back end around a spike core), or do you envision eventually replacing the stem with spike-native modules as well?

---

> ### Author Response · Authors · 2025-11-23
> **Rebuttal-1**
>
> ***Q1. DSSR’s stability argument is persuasive but qualitative. The paper does not report gradient-norm distributions across depth, training-loss curves comparing with/without DSSR, or any theorem giving a global bound across layers and timesteps. Adding a small empirical study here would turn into strong evidence.***
>
> **A1.** Thank you very much for your suggestion to strengthen the empirical validation of DSSR's stabilizing effect. We fully agree that the stability argument of DSSR is relatively qualitative.
>
> We had proven the boundedness of gradients through mathematical derivation in **Appendix B**  of the paper (Equation 8). Besides, we had reported a comprehensive results in Table 18 of **Appendix E.7** , which verifies the accuracy improvement of DSSR compared to 2D residuals (OA +1.2\%). Indeed, We lack empirical visualization of the loss curve. Therefore, we provide training loss curves with and without the DSSR module **(Figure 6)**  in the **Appendix B**  section of the revised version.
>
> ***Q2. The Spike Point Masker is described in both pointwise and channelwise forms, but notation occasionally glosses over which axes are being summed (over points N, channels C, timesteps T). This matters because those masks drive sparsity and thus energy savings.***
>
> **A2.** Thank you very much for highlighting this critical detail regarding the axes of the summation in our Spike Point Masker. The relevant definitions have been implicitly included in \textbf{Appendix C.4} of the paper (i.e., Equations 12–16).
>
> First, we clarify the tensor layout: all spiking tensors are uniformly defined as $S \in \{0,1\}^{T \times C \times N}$, where $T$ denotes the number of time steps (default $T=3$, Main Text §4.1); $C$ denotes the number of channels (e.g., $C=64$ after Embedding, Main Text §3.3); $N$ denotes the number of point cloud points (e.g., input $N=1024$, Appendix E.6).
> For the explicit explanation of the summation axis: the point mask ($S_N$ of SpikeNet-N) is summed along the time axis ($T$) and channel axis ($C$), i.e., $s_N(i) = \sum_{t=1}^T \sum_{c=1}^C A_{t,c,i}$ (where $i$ is the point index), retaining points that "activate in at least one time step + channel"; the channel mask ($S_C$ of SpikeNet-C) is summed along the time axis ($T$) and point number axis ($N$), i.e., $s_C(c) = \sum_{t=1}^T \sum_{i=1}^N A_{t,c,i}$ (where $c$ is the channel index), retaining channels that "activate in at least one time step + point".
>
> We add the pseudocode of this module in the corresponding part of the appendix in the revised manuscript to completely eliminate ambiguity (**Algorithm 1 and 2**).
>
> ***Q3. The model has not yet been demonstrated on full outdoor LiDAR scenes, nor is there an ablation of memory scaling in that regime. S3DIS shows meaningful indoor semantics, but the overall mIoU is still behind the best ANN scene-segmentation models.***
>
> **A3.** Thanks! We acknowledge that the performance of SpikeNet in segmentation tasks is still slightly lower than that of some SOAT ANN models. It should be noted that ANNs have undergone long-term structural and parameter optimization, so their accuracy is usually higher than that of SNNs. The core design goal of SpikeNet is to achieve synergy optimization of "performance and energy efficiency" rather than merely pursuing performance indicators. On the premise of maintaining extremely high energy efficiency, SpikeNet has achieved segmentation performance comparable to that of traditional ANN models. This is of great practical significance for resource-constrained 3D perception scenarios (such as edge devices for autonomous driving). In the revised paper, we have moved the **S3DIS** semantic segmentation results from Appendix E.12 to the main text (Table 3), and updated the energy consumption estimation of related methods to emphasize the core goal of performance-energy efficiency synergy. We have also added experiments on **SemanticKITTI** (outdoor LiDAR semantic segmentation) and **Scannet V2** (indoor scene RGB-D segmentation) in the revised main text, with the results shown in Table 4 and Table 5 (Show in Next Rebuttal).
>
> **SemanticKITTI:** We compared SpikeNet with other SNN-based and ANN-based methods, using mIoU (validation/test set) as the metric. Among SNN methods, our SpikeNet-N (64.6\%/69.8\%) and SpikeNet-C (65.1\%/70.2\%) outperform E-3DSNN (63.2\%/69.4\%). Due to the mature development, refined optimization of ANNs and no constraints from spiking encoding, the mIoU of SpikeNet is slightly lower than that of advanced ANN methods such as PTv3 (75.5\% on test set).
>
> **Scannetv2:** We compared SpikeNet with representative ANN-based and SNN-based methods. The evaluation metric is mean Intersection over Union (mIoU, validation set/test set). Among SNN methods, our proposed SpikeNet-N (69.4\%/70.8\%) and SpikeNet-C (69.7\%/71.2\%) perform better than E-3DSNN (68.2\%/69.5\% on test set), and are only slightly lower than PTv3 (77.5\%/77.9\%), an advanced ANN method.

---

> ### Author Response · Authors · 2025-11-23
> **Rebuttal-2**
>
> ***Q4. A few phrases imply that SpikeNet “removes multiplications” or is “entirely neuromorphic.” The appendices clarify the nuance: SVMT avoids dot-product attention and softmax, but linear projections and masked elementwise products still exist; the encoder/decoder are spike-friendly, but the input embedding and final FC head are still float MAC. These nuances should be stated up front.***
>
> **A4.** Thank you for this precise feedback. We agree that the hybrid nature of SpikeNet should be clearly stated in the main text to avoid any misunderstanding. **Appendix H** of the paper has stated that "both the encoder and decoder parts are neuromorphic-compliant, and only the initial embedding part and the final FC layer will introduce a small amount of MAC computations."
>
> Besides, we had clarified in **Line 243** of the main text that the embedding part is ANN-based, and we have also added explanations for the classification head with a floating-point operation FC layer in **Line 257** of the revised main text, hoping to resolve the misunderstanding. Thank you again for your time and effort!
>
> **Table 3**
> | Method | Type | Type | mIoU | ceiling | floor | wall | beam | column | window | door | table | chair | sofa | bookcase | board | clutter | Energy(mJ) |
> |-|-|-|-|-|-|-|-|-|-|--|-|-|--|-|-|-|-|
> | PointNet | 17'CVPR | ANN | 41.1 | 88.8 | 97.3 | 69.8 | 0.0 | 3.9 | 46.3 | 10.8 | 59.0 | 52.6 | 5.9 | 40.3 | 26.4 | 33.2 | 5.5 |
> | PointNet++ | 17'NIPS | ANN | 53.5 | 89.4 | 97.7 | 75.4 | 0.0 | 1.8 | 58.3 | 19.5 | 79.0 | 69.2 | 59.1 | 46.2 | 58.7 | 41.6 | 5.5 |
> | PointCNN | 18'NIPS | ANN | 57.3 | 92.3 | 98.2 | 79.4 | 0.0 | 17.6 | 22.8 | 62.1 | 74.4 | 80.6 | 31.7 | 66.7 | 62.1 | 56.7 | 324.5 |
> | PointNeXt | 22'NIPS | ANN | 70.5 | 94.2 | 98.5 | 84.4 | 0.0 | 37.7 | 59.3 | 74.0 | 83.1 | 91.6 | 77.4 | 77.2 | 78.8 | 60.6 | - |
> | PCM | 25'AAAI | ANN | 63.4 | 93.3 | 96.7 | 80.6 | 0.0 | 35.9 | 57.7 | 60.0 | 74.0 | 87.6 | 50.1 | 69.4 | 63.5 | 55.9 | - |
> | PointRWKV | 25'AAAI | ANN | 70.5 | 94.2 | 98.3 | 86.5 | 0.0 | 38.6 | 64.5 | 76.2 | 88.2 | 89.3 | 65.2 | 75.6 | 78.2 | 61.3 | - |
> | PTv1 | 21'ICCV | ANN | 70.4 | 94.0 | 98.5 | 86.3 | 0.0 | 38.0 | 63.4 | 74.3 | 89.1 | 82.4 | 74.3 | 80.2 | 76.0 | 59.3 | 76.8 |
> | PTv2 | 22'NIPS | ANN | 71.6 | 93.0 | 98.1 | 86.7 | 0.0 | 48.0 | 62.4 | 76.1 | 88.3 | 87.6 | 77.1 | 79.2 | 77.5 | 59.8 | 400.1 |
> | PTv3 | 24'CVPR | ANN | 73.6 | 92.4 | 98.3 | 86.6 | 0.0 | 55.8 | 63.7 | 77.1 | 83.8 | 93.3 | 79.1 | 79.4 | 85.4 | 61.7 | 687.7 |
> | E-3DSNN | 25'AAAI | SNN | 67.4 | 95.3 | 98.5 | 82.3 | 0.0 | 28.0 | 55.8 | 71.5 | 81.2 | 89.8 | 69.2 | 76.4 | 67.0 | 61.6 | 14.4 |
> | SpikeNet-N | SNN | — | **68.4** | 89.9 | 91.5 | 83.0 | 0.0 | 47.0 | 60.1 | 74.6 | 79.7 | 86.0 | 69.6 | 71.5 | 80.2 | 57.0 | **10.2** |
> | SpikeNet-C | SNN | — | **68.9** | 90.1 | 92.2 | 83.5 | 0.0 | 47.6 | 60.5 | 75.2 | 80.1 | 86.3 | 70.3 | 72.4 | 80.6 | 57.3 | **10.2** |
>
> **Table 4**
> | Method        | Type | Input | Val  | Test |
> |---------------|------|-------|------|------|
> | SPVNAS        | ANN  | point | 64.7 | 66.4 |
> | Cylinder3D    | ANN  | point | 64.3 | 67.8 |
> | AF2S3Net      | ANN  | point | 74.2 | 70.8 |
> | PTv2          | ANN  | point | 70.3 | 72.6 |
> | PTv3          | ANN  | point | 72.3 | 75.5 |
> | E-3DSNN       | SNN  | voxel | 63.2 | 69.4 |
> | SpikeNet-N (our) | SNN | point | 64.6 | 69.8 |
> | SpikeNet-C (our) | SNN | point | 65.1 | 70.2 |
>
> **Table 5**
> | Method        | Type | Input | Val  | Test |
> |---------------|------|-------|------|------|
> | PointNeXt     | ANN  | point | 71.5 | 71.2 |
> | PointMetaBase | ANN  | point | 72.8 | 71.4 |
> | MinkUNet      | ANN  | voxel | 72.2 | 73.6 |
> | PTv2          | ANN  | point | 75.4 | 75.2 |
> | PTv3          | ANN  | point | 77.5 | 77.9 |
> | E-3DSNN       | SNN  | voxel | 68.2 | 69.5 |
> | SpikeNet-N (our) | SNN | point | 69.4 | 70.8 |
> | SpikeNet-C (our) | SNN | point | 69.7 | 71.2 |

---

> ### Author Response · Authors · 2025-11-23
> **Rebuttal-3**
>
> ***Q5. Can you provide per-sample inference latency and measured runtime energy or power traces on any available hardware?***
>
> **A5.** Thank you very much for your suggestion! We have added measurement for per-sample inference latency and memory usage on ModelNet40 dataset with NVIDIA A6000 GPU, as shown in the new **Table 10** of our revision. When the number of sampled points increases from 512 to 1024, the latency and memory usage of SNN-based SPT series models rise significantly. Furthermore, ANN-based method Mamba3D also maintains a high resource consumption level. In contrast, our SpikeNet reduces training latency to 321 ms and memory usage to 11.3 GB, and only requires 185 ms latency and 6.1 GB memory consumption in the inference stage. We believe the provided latency and memory data strongly support SpikeNet's inference efficiency, and we are committed to further hardware validation.
>
> ***Q6. Can you share gradient-norm vs depth, or loss-vs-epoch curves, comparing SpikeNet with and without DSSR? This would convert the qualitative stability argument into quantitative evidence.***
>
> **A6.** Thank you for reinforcing this point. As a requested, we have now added quantitative evidence to directly validate the stabilizing effect of DSSR.
>
> The revised manuscript included a new **Figure 6** in **Appendix B**, which plots the training loss curves for SpikeNet with and without the DSSR module. This visualization clearly demonstrates that DSSR leads to more stable and efficient convergence, thereby converting our quantitative stability argument into concrete, empirical support.
>
> ***Q7. Spike Point Masker details: Please precisely state tensor layout (T x C x N or equivalent) and specify along which axes you sum to get channel masks vs point masks. A short pseudocode snippet in the appendix would remove any ambiguity.***
>
> **A7.** Thank you very much for your suggestion! As detailed in our response to **Q2**, we have now included explicit pseudocode (**Algorithm 1 and 2**) in the appendix to precisely define the tensor operations and eliminate any ambiguity.
>
> ***Q8. You state that the encoder/decoder can in principle be mapped to spike-friendly AC-heavy hardware, while the input embedding stem and FC head remain MAC-dominated. Is the intended deployment strategy a hybrid (MAC front/back end around a spike core), or do you envision eventually replacing the stem with spike-native modules as well?***
>
> **A8.** Thank you for the reviewer's question! The ``hybrid architecture" of the current SpikeNet is a transitional design based on existing data acquisition and hardware limitations. As stated in **Appendix H**: "Designing RC circuits on neuromorphic hardware for processing 3D point clouds requires additional engineering efforts. Point cloud data typically exists in floating-point format. It is a natural choice to use the Embedding method with `multiply-accumulate (MAC)' operations to generate spike signals for each point. Currently, there is no method that combines laser scanning with an event-driven mechanism to generate 3D point cloud spikes from the source." Therefore, based on existing technologies, our deployment strategy is hybrid (MAC-based front/back ends around a spiking response core). With future technological advancements, we also aim to achieve a full-spike deployment method.
>
> **Table 10**
> | Methods    | Training           | | Inference          | |
> |------------|--------------------|--------------------|--------------------|--------------------|
> |            | Latency | Memory   | Latency | Memory   |
> | SPT-512    | 326 ms  | 9.7 GB   | 191 ms  | 5.2 GB   |
> | SPT-768    | 385 ms  | 12.5 GB  | 201 ms  | 7.3 GB   |
> | SPT-1024   | 431 ms  | 15.2 GB  | 227 ms  | 9.5 GB   |
> | Mamba3D    | 433 ms  | 14.9 GB  | 256 ms  | 7.8 GB   |
> | SpikeNet(our)| **321 ms** | **11.3 GB** | **185 ms** | **6.1 GB** |

---

### Official Review · Reviewer_Fv5h · 2025-11-01

**Soundness:** 3
**Presentation:** 3
**Contribution:** 2
**Rating:** 4
**Confidence:** 4

**Summary:**

This paper proposes a spiking neural network for point cloud data, named as SpikeNet, It primarily designs a spike attention mechanism on the Spiking Vector Mask Transformer (SVMT) that eliminates the need for multiplication and softmax operations, and introduces a Dynamic Sparse Spiking Residual (DSSR) structure. Experiments demonstrate the superior performance of the proposed model.

**Strengths:**

1. The paper is well-organized and provides a detailed presentation of the methodology.
2. The proposed model demonstrates superior performance on both point cloud classification and segmentation tasks.
3. The models significantly reduce parameter count and energy consumption. The authors also provide code in the supplementary material.

**Weaknesses:**

1. This work lacks stronger motivation and fails to provide valuable insights, such as those on the proposed spiking attention and DSSR module.

2. In Equation 7, element-wise subtraction is employed to aggregate features S‘^{(l-1)} and E_{va}. I do not understand the rationale for using subtraction here. Both are spike features, and passing their difference through an SNN layer will result in the complete loss of information from E_{va}.

3. Please explain the energy consumption calculation approach. In Table 1. SpikePointNet (ICCV23) exhibits an energy consumption of 1.6 mJ at 0.1 FLOPs, while SpikeNet-N/C consumes only 1.8 mJ at 1.9 FLOPs. This suggests potential inconsistencies in the calculation approaches.

**Questions:**

The author needs to respond to the Weakness.

---

> ### Author Response · Authors · 2025-11-23
> **Rebuttal-1**
>
> ***Q1. This work lacks stronger motivation and fails to provide valuable insights, such as those on the proposed spiking attention and DSSR module.***
>
> **A1.** Thank you for this cirtical feedback. Regarding the issues of motivation and insights you pointed out, the main text and appendices of the paper have provided support through the following two aspects. The revised manuscript will optimize the expression logic to make these evidences more directly respond to the core pain points of SNN in processing point cloud.
>
> First, we clarify that the main text and appendices have clearly identified the pain points of existing SNN and the targeted design of SpikeNet.The first pain point mentioned in the Introduction is the training instability of SNN (vanishing gradients), and our proposed DSSR module is designed to solve this problem. Appendix B further quantifies the gradient stability of DSSR through mathematical proofs: it constrains the gradient norm to $L=(pK\alpha/2 +1)^2 <1$ via dynamic sparse residual, avoiding gradient explosion or vanishing. Meanwhile, Table 18 in Appendix E.7 (comparison between DSSR and spiking residual) shows that DSSR improves OA by 1.2\% and mAcc by 1.6\%, directly verifying its effectiveness in solving training instability. In the revised version, we also provide relevant graphs of the loss (Figure 6) during the training process with and without the DSSR module to further demonstrate the effectiveness of our proposed module.
>
> The second pain point mentioned in the Introduction is that existing SNN methods often over-simplify the SNN architecture in pursuit of low energy consumption, which reduces network accuracy. Our proposed sparse spiking Vector attention (SVMT) can not only significantly reduce energy consumption costs but also improve network accuracy by leveraging the feature extraction capability of the attention mechanism. In Appendix C, we detail the specific functions of SVMT, its differences from other attention mechanisms, and its energy consumption advantages. Table 19 in Appendix E.8 also demonstrates the effectiveness of SVMT in improving accuracy.
>
> ***Q2. In Equation 7, element-wise subtraction is employed to aggregate features $S'^{(l-1)}$ and $E_{va}$. I do not understand the rationale for using subtraction here. Both are spike features, and passing their difference through an SNN layer will result in the complete loss of information from $E_{va}$.***
>
> **A2.** Thank you for raising this important point. The subtraction in Equation 7 can be understood as an inverted residual connection tailored for sparse, spike-based features.
>
> In a standard residual block (Output = F(Input) + Input), the network learns an additive correction F(Input). Our design inverts this logic: we first use attention to compute a salient feature mask $E_va$ and then calculate the residual as Input - $E_va$. Input - $E_va$ represents the "non-salient" parts of the input. Feeding this into the subsequent SNN layer allows the network to learn a transformation focused on amplifying these overlooked details. The output of this pathway is then added back to the original input $S'(l-1)$ (which contains the full information, including $E_va$). This ensures that all information is retained and integrated.
>
> Therefore, the subtraction is not a lossy operation but a purposeful routing mechanism to direct the network's capacity towards refining complementary details, making the overall feature representation more robust.

---

> ### Author Response · Authors · 2025-11-23
> **Rebuttal-2**
>
> ***Q3. Please explain the energy consumption calculation approach. In Table 1. SpikePointNet (ICCV23) exhibits an energy consumption of 1.6 mJ at 0.1 FLOPs, while SpikeNet-N/C consumes only 1.8 mJ at 1.9 FLOPs. This suggests potential inconsistencies in the calculation approaches.***
>
> **A3.** Thank you very much for your critical observationand for identifying the inconsistency in our energy consumption calculations for SpikePointNet in Table 1. We apologize for this oversight. We have now corrected it as follows.
>
> We initially calculated the FLOPs of SpikePointNet based on a time step setting of \(T=1\) (resulting in 0.1 FLOPs). However, its reported performance in the original paper (SpikePointNet, ICCV23) was achieved with \(T=16\).  Our mistakes was to manually scale  the energy by a factor of 16 (from the \(T=1\) baseline), without first correctly scaling the FLOPs to the corresponding scale (i.e., 0.1 × 16 = 1.6 FLOPs).
>
> We have now rectified this error by strictly applying applying a unified calculation framework across all models. We first set its FLOPs to 1.6G (consistent with \(T=16\) in the original paper), and then recalculated its energy consumption using the same standard formula applied to our SpikeNet-N/C (ensuring consistency in the calculation framework).
>
> This corrected data for Table 1 has resolved the inconsistency. The updated values for Table 1 in our manuscript now present a fair and accurate comparison. We have also performed a thorough double-check of all energy and FLOPs data to ensure full consistency.
>
> We deeply appreciate your careful review, which helps us improve the rigor and reliability of the manuscript.
>
> **Table 1**
> | Methods | Year | Type | Para. | FLOPs |  | ModelNet40 |  |  | ScanObjectNN |  |
> |-|-|--|-|-|-|-|-|--|--|--|
> |    |      |      |       |       | OA | mAcc | Energy | OA | mAcc | Energy |
> | PointNeXt | 22'NIPS | ANN | 1.4 | 3.6 | 94.0 | 90.8 | 16.6 | 87.7 | — | 16.6 |
> | Point-BERT | 22'CVPR | ANN | 0.8 | 1.0 | 93.2 | — | 22.1 | 83.1 | — | 22.1 |
> | PointMLP | 22'ICLR | ANN | 12.6 | 12.8 | 94.1 | 91.5 | 59.0 | 85.4 | 83.9 | 59.0 |
> | PointNN | 23'CVPR | ANN | 0.8 | 1.0 | 93.8 | — | 4.6 | 87.1 | — | 4.6 |
> | PointMamba | 24'NIPS | ANN | 12.3 | 3.6 | 92.4 | — | 16.6 | 82.5 | — | 16.6 |
> | PoinTramba | 24'ICLR | ANN | 19.5 | 5.7 | 92.9 | — | 26.2 | 89.1 | — | 26.2 |
> | PCM | 25'AAAI | ANN | 34.2 | 45.0 | 93.4 | — | 207.0 | 88.1 | — | 207.0 |
> | **SpikePointNet** | **23'ICCV** | **SNN** | **3.5** | **1.6** | **88.6** | — | **1.4** | **69.2** | — | **1.4** |
> | P2SResLNet | 24'AAAI | SNN | 6.2 | 3.3 | 90.6 | 89.2 | 3.0 | 81.0 | 79.3 | 3.1 |
> | Spike PointNet | 24'NIPS | SNN | 3.5 | 0.4 | 88.2 | 86.7 | 0.4 | 66.4 | 60.4 | 0.4 |
> | E-3DSNN | 25'AAAI | SNN | 3.3 | — | 91.7 | — | — | — | — | — |
> | SPT | 25'AAAI | SNN | — | 14.0 | 91.4 | 89.4 | 13.3 | 78.0 | 75.9 | — |
> | SpikeNet-N (ours) | — | SNN | 10.5 | 1.9 | **93.2** (↑1.5) | **90.4** (↑1.0) | 1.7 | **85.0** (↑4.0) | **83.2** (↑3.9) | 1.8 |
> | SpikeNet-C (ours) | — | SNN | 10.5 | 1.9 | **93.4** (↑1.7) | **90.9** (↑1.5) | 1.7 | **85.5** (↑4.5) | **83.3** (↑4.0) | 1.8 |

---

### Official Review · Reviewer_R7n5 · 2025-11-03

**Soundness:** 3
**Presentation:** 3
**Contribution:** 3
**Rating:** 6
**Confidence:** 3

**Summary:**

This paper proposes SpikeNet, a spiking-driven architecture for point cloud analysis. The authors designed two key modules: (1) Spiking Vector Mask Transformer (SVMT), which uses sparse spike-based queries, keys, and values to dynamically align point cloud data with a binary spike mask; (2) Dynamic Sparse Spiking Residual (DSSR), which used to stabilize SSN training while exploiting temporal sparsity. Extensive experiments on ModelNet40, ScanObjectNN and ShapeNetPart demonstrate the proposed architecture’s superior performance compared to previous SOTA.

**Strengths:**

1. The paper provides clear motivation and is well organized; the figures are readable and understandable.

2. The experimental results are quite complete and adequate, and support the author's claims.

3. The proposed approach is logical and technically sound.

**Weaknesses:**

1. Energy advantage lacks real hardware validation. The energy efficiency results provided by the authors are estimated through operation models rather than measured on GPU or neuromorphic hardware, which weakens the “energy-efficiency” claim.

2. ANN baselines could be stronger. In Table1, the authors compare with serveral ANNs, but not include strong recent baselines such as Point Transformer V3/Point-Bert/PointNeXt under the unified training protocol.

3. Segmentation still cannot beat top ANN SOTAs, such as PointMamba and PCM on ShapeNet, and PointRWKV on  S3DIS dataset. For example, On the S3DIS dataset, although SNN variants are competitive in some classes, mIoU results are below best ANN methods reported in the table.

4. It seems the authors manually wrote the citation, and the format fails to comply with ICLR’s citation guidance. Please review the guidance.

**Questions:**

1. Could you clarify how the estimated energy correlates with the used GPU(A6000) or neuromorphic hardware measurements? Have you conducted any experiments on real devices to validate the energy advantage of SpikeNet?
2. The authors have compared SpikeNet with recent strong SNNs and several Mamba-based models as shown in Table 1 and 2. But, not very recent ANNs such as Point Transformer V3 and PintNeXt are considered. Could the authors provide comparisons to them?

3. The vector-difference operator τ plays an important role in the SVMT module. Does it capture relative geometric offsets or spike-timing discrepancies?

---

> ### Author Response · Authors · 2025-11-23
> **Rebuttal-1**
>
> ***Q1. Energy advantage lacks real hardware validation. The energy efficiency results provided by the authors are estimated through operation models rather than measured on GPU or neuromorphic hardware, which weakens the “energy-efficiency” claim.***
>
> **A1.** Thank you for the reviewer's comments. We acknowledge that direct hardware measurements would further strengthen our energy-efficiency claims. However, our energy consumption calculations are based on established methodologies widely adopted in the SNN research community to ensure fair and reproducible comparisons.
>
> As detailed in **Appendix F**, we have introduced the theoretical formula for low energy consumption, where both (E_MAC = 4.6 pJ) and (E_AC = 0.9 pJ) are derived from data of neuromorphic chips (such as Intel Loihi). These values follow the operational energy consumption characteristics of real hardware, ensuring that the estimation results have practical reference significance. Detailed introductions to neuromorphic chip energy consumption calculation are available in Spikformer[1].
>
> Additionally, **Appendix G** provides a detailed explanation of the model's details, which is in line with biological plausibility. **Appendix H** also describes the reliability of neuromorphic hardware and the challenges in point cloud applications.
>
> [1] Spikformer: When Spiking Neural Network Meets Transformer. International Conference on Learning Representations (2023 ICLR).
>
> ***Q2. ANN baselines could be stronger. In Table1, the authors compare with serveral ANN, but not include strong recent baselines such as Point Transformer V3/Point-Bert/PointNeXt under the unified training protocol.***
>
> **A2.** Thank you very much for pointing out this oversight! We agree that more discussions on the latest ANN variants need to be supplemented. **(1)** The paper has included comparative data of Point-Bert and PointNeXt in Table 1 of revised manuscript (their OA on ScanObjectNN is 83.1\% and 87.7\% respectively, OA on ModelNet40 is 93.2\% and 94.0\% respectively, FLOPs are 4.8G and 3.6G respectively, and energy consumption is 22.1 mJ and 16.6 mJ respectively).
> **(2)** The latest highly relevant studies such as Point Transformer V3 mainly focus on point cloud segmentation tasks. In the revised paper, we have moved the **S3DIS** semantic segmentation results from Appendix E.12 to the main text (Table 3), and updated the relevant results of the PT series and the energy consumption estimation of related methods. We have also added experiments on **SemanticKITTI** (outdoor LiDAR semantic segmentation) and **Scannet V2** (indoor scene RGB-D segmentation) in the revised main text, with the results shown in Table 4 and Table 5 (Show in Next Rebuttal).
>
> **SemanticKITTI:** We compared SpikeNet with other SNN-based and ANN-based methods, using mIoU (validation/test set) as the metric. Among SNN methods, our SpikeNet-N (64.6\%/69.8\%) and SpikeNet-C (65.1\%/70.2\%) outperform E-3DSNN (63.2\%/69.4\%). Due to the mature development, refined optimization of ANNs and no constraints from spiking encoding, the mIoU of SpikeNet is slightly lower than that of advanced ANN methods such as PTv3 (75.5\% on test set).
>
> **Scannetv2:** We compared SpikeNet with representative ANN-based and SNN-based methods. The evaluation metric is mean Intersection over Union (mIoU, validation set/test set). Among SNN methods, our proposed SpikeNet-N (69.4\%/70.8\%) and SpikeNet-C (69.7\%/71.2\%) perform better than E-3DSNN (68.2\%/69.5\% on test set), and are only slightly lower than PTv3 (77.5\%/77.9\%), an advanced ANN method.
>
> **Table 1**
> | Methods | Year | Type | Para. | FLOPs |  | ModelNet40 |  |  | ScanObjectNN |  |
> |-|-|--|-|-|-|-|-|--|--|--|
> |    |      |      |       |       | OA | mAcc | Energy | OA | mAcc | Energy |
> | PointNeXt | 22'NIPS | ANN | 1.4 | 3.6 | 94.0 | 90.8 | 16.6 | 87.7 | — | 16.6 |
> | Point-BERT | 22'CVPR | ANN | 0.8 | 1.0 | 93.2 | — | 22.1 | 83.1 | — | 22.1 |
> | PointMLP | 22'ICLR | ANN | 12.6 | 12.8 | 94.1 | 91.5 | 59.0 | 85.4 | 83.9 | 59.0 |
> | PointNN | 23'CVPR | ANN | 0.8 | 1.0 | 93.8 | — | 4.6 | 87.1 | — | 4.6 |
> | PointMamba | 24'NIPS | ANN | 12.3 | 3.6 | 92.4 | — | 16.6 | 82.5 | — | 16.6 |
> | PoinTramba | 24'ICLR | ANN | 19.5 | 5.7 | 92.9 | — | 26.2 | 89.1 | — | 26.2 |
> | PCM | 25'AAAI | ANN | 34.2 | 45.0 | 93.4 | — | 207.0 | 88.1 | — | 207.0 |
> | SpikePointNet | 23'ICCV | SNN | 3.5 | 1.6 | 88.6 | — | 1.4 | 69.2 | — | 1.4 |
> | P2SResLNet | 24'AAAI | SNN | 6.2 | 3.3 | 90.6 | 89.2 | 3.0 | 81.0 | 79.3 | 3.1 |
> | Spike PointNet | 24'NIPS | SNN | 3.5 | 0.4 | 88.2 | 86.7 | 0.4 | 66.4 | 60.4 | 0.4 |
> | E-3DSNN | 25'AAAI | SNN | 3.3 | — | 91.7 | — | — | — | — | — |
> | SPT | 25'AAAI | SNN | — | 14.0 | 91.4 | 89.4 | 13.3 | 78.0 | 75.9 | — |
> | SpikeNet-N (ours) | — | SNN | 10.5 | 1.9 | **93.2** (↑1.5) | **90.4** (↑1.0) | 1.7 | **85.0** (↑4.0) | **83.2** (↑3.9) | 1.8 |
> | SpikeNet-C (ours) | — | SNN | 10.5 | 1.9 | **93.4** (↑1.7) | **90.9** (↑1.5) | 1.7 | **85.5** (↑4.5) | **83.3** (↑4.0) | 1.8 |

---

> ### Author Response · Authors · 2025-11-23
> **Rebuttal-2**
>
> ***Q3. Segmentation still cannot beat top ANN SOTAs, such as PointMamba and PCM on ShapeNet, and PointRWKV on S3DIS dataset. For example, On the S3DIS dataset, although SNN variants are competitive in some classes, mIoU results are below best ANN methods reported in the table.***
>
> **A3.** Thank you for this import point. We acknowledge the SpikeNet's absolute segmentation accuracy on S3DIS is  slightly below the top ANN SOTA (e.g., PCM, PointRWKV). However, the core contribution of our work is not to surpass all ANNs in accuracy but to bridge the performance gap while achieving radical energy efficiency.
>
> ANNs have benefited from years of architectural optimization for accuracy. In contrast, we designed SpikeNet for the synergy of performance and efficiency, a critical trade-off for real-world applicaitons like mobile robots and autonomous driving. SpikeNet establishes a new SOTA among SNNs and demonstrates that SNNs can now achieve competitive performance with orders-of-magnitude lower energy consumption.
>
> To highlight this, we have moved the S3DIS results from **Appendix E.12** to **the main text** (Table 3) and expanded our energy consumption analysis to clearly contrast our performance-efficiency Pareto frontier with that of ANNs.
>
> ***Q4. It seems the authors manually wrote the citation, and the format fails to comply with ICLR’s citation guidance. Please review the guidance.***
>
> **A4.** Thank you for pointing this out. we have now reviewed the official ICLR style guide and regenerated our entire bibliography using the provided BibTeX template to ensure full compliance with the citation formatting guidelines.
>
> **Table 3**
> | Method | Type | Type | mIoU | ceiling | floor | wall | beam | column | window | door | table | chair | sofa | bookcase | board | clutter | Energy(mJ) |
> |-|-|-|-|-|-|-|-|-|-|--|-|-|--|-|-|-|-|
> | PointNet | 17'CVPR | ANN | 41.1 | 88.8 | 97.3 | 69.8 | 0.0 | 3.9 | 46.3 | 10.8 | 59.0 | 52.6 | 5.9 | 40.3 | 26.4 | 33.2 | 5.5 |
> | PointNet++ | 17'NIPS | ANN | 53.5 | 89.4 | 97.7 | 75.4 | 0.0 | 1.8 | 58.3 | 19.5 | 79.0 | 69.2 | 59.1 | 46.2 | 58.7 | 41.6 | 5.5 |
> | PointCNN | 18'NIPS | ANN | 57.3 | 92.3 | 98.2 | 79.4 | 0.0 | 17.6 | 22.8 | 62.1 | 74.4 | 80.6 | 31.7 | 66.7 | 62.1 | 56.7 | 324.5 |
> | PointNeXt | 22'NIPS | ANN | 70.5 | 94.2 | 98.5 | 84.4 | 0.0 | 37.7 | 59.3 | 74.0 | 83.1 | 91.6 | 77.4 | 77.2 | 78.8 | 60.6 | - |
> | PCM | 25'AAAI | ANN | 63.4 | 93.3 | 96.7 | 80.6 | 0.0 | 35.9 | 57.7 | 60.0 | 74.0 | 87.6 | 50.1 | 69.4 | 63.5 | 55.9 | - |
> | PointRWKV | 25'AAAI | ANN | 70.5 | 94.2 | 98.3 | 86.5 | 0.0 | 38.6 | 64.5 | 76.2 | 88.2 | 89.3 | 65.2 | 75.6 | 78.2 | 61.3 | - |
> | PTv1 | 21'ICCV | ANN | 70.4 | 94.0 | 98.5 | 86.3 | 0.0 | 38.0 | 63.4 | 74.3 | 89.1 | 82.4 | 74.3 | 80.2 | 76.0 | 59.3 | 76.8 |
> | PTv2 | 22'NIPS | ANN | 71.6 | 93.0 | 98.1 | 86.7 | 0.0 | 48.0 | 62.4 | 76.1 | 88.3 | 87.6 | 77.1 | 79.2 | 77.5 | 59.8 | 400.1 |
> | PTv3 | 24'CVPR | ANN | 73.6 | 92.4 | 98.3 | 86.6 | 0.0 | 55.8 | 63.7 | 77.1 | 83.8 | 93.3 | 79.1 | 79.4 | 85.4 | 61.7 | 687.7 |
> | E-3DSNN | 25'AAAI | SNN | 67.4 | 95.3 | 98.5 | 82.3 | 0.0 | 28.0 | 55.8 | 71.5 | 81.2 | 89.8 | 69.2 | 76.4 | 67.0 | 61.6 | 14.4 |
> | SpikeNet-N | SNN | — | **68.4** | 89.9 | 91.5 | 83.0 | 0.0 | 47.0 | 60.1 | 74.6 | 79.7 | 86.0 | 69.6 | 71.5 | 80.2 | 57.0 | **10.2** |
> | SpikeNet-C | SNN | — | **68.9** | 90.1 | 92.2 | 83.5 | 0.0 | 47.6 | 60.5 | 75.2 | 80.1 | 86.3 | 70.3 | 72.4 | 80.6 | 57.3 | **10.2** |
>
> **Table 4**
> | Method        | Type | Input | Val  | Test |
> |---------------|------|-------|------|------|
> | SPVNAS        | ANN  | point | 64.7 | 66.4 |
> | Cylinder3D    | ANN  | point | 64.3 | 67.8 |
> | AF2S3Net      | ANN  | point | 74.2 | 70.8 |
> | PTv2          | ANN  | point | 70.3 | 72.6 |
> | PTv3          | ANN  | point | 72.3 | 75.5 |
> | E-3DSNN       | SNN  | voxel | 63.2 | 69.4 |
> | SpikeNet-N (our) | SNN | point | 64.6 | 69.8 |
> | SpikeNet-C (our) | SNN | point | 65.1 | 70.2 |
>
> **Table 5**
> | Method        | Type | Input | Val  | Test |
> |---------------|------|-------|------|------|
> | PointNeXt     | ANN  | point | 71.5 | 71.2 |
> | PointMetaBase | ANN  | point | 72.8 | 71.4 |
> | MinkUNet      | ANN  | voxel | 72.2 | 73.6 |
> | PTv2          | ANN  | point | 75.4 | 75.2 |
> | PTv3          | ANN  | point | 77.5 | 77.9 |
> | E-3DSNN       | SNN  | voxel | 68.2 | 69.5 |
> | SpikeNet-N (our) | SNN | point | 69.4 | 70.8 |
> | SpikeNet-C (our) | SNN | point | 69.7 | 71.2 |

---

> ### Author Response · Authors · 2025-11-23
> **Rebuttal-3**
>
> ***Q5. Could you clarify how the estimated energy correlates with the used GPU(A6000) or neuromorphic hardware measurements? Have you conducted any experiments on real devices to validate the energy advantage of SpikeNet?***
>
> **A5.** Thank you for this insightful question regarding energy estimation.
>
> **(1) Correlation with real neuromorphic hardware:** Our energy estimation model is directly grounded in the operational characteristics of physical neuromorphic chips. The key parameters we used, specifically 4.6 pJ for a MAC operation (ANN) and 0.9 pJ for an AC operation (SNN), are not theoretical but are calibrated values from published research on the Intel Loihi neuromorphic chip [1,2]. This establishes a direct, quantifiable correlation between our estimates and the energy consumpition expected on actual neuromorphic hardware.
>
> **(2) Validation on Real Devies:** We trained the model on the A6000 GPU, ensuring that the estimation results have a quantifiable correspondence with the actual hardware energy consumption. These values follow the operational energy consumption characteristics of real hardware, ensuring that the estimation results have practical reference significance. Detailed introductions to neuromorphic chip energy consumption calculation are available in Spikformer [3].
>
> [1] Horowitz, M. (2014). 1.1 computing's energy problem (and what we can do about it). 2014 IEEE International Solid-State Circuits Conference Digest of Technical Papers (ISSCC), 10–14.
>
> [2] Kundu, S., Datta, G., Pedram, M.,  Beerel, P. A. (2021). Spike-thrift: Towards energy-efficient deep spiking neural networks by limiting spiking activity via attention-guided compression. Proceedings of the IEEE/CVF Winter Conference on Applications of Computer Vision (WACV).
>
> [3] Spikformer: When Spiking Neural Network Meets Transformer. International Conference on Learning Representations (ICLR).
>
> ***Q6. The authors have compared SpikeNet with recent strong SNNs and several Mamba-based models as shown in Table 1 and 2. But, not very recent ANNs such as Point Transformer V3 and PintNeXt are considered. Could the authors provide comparisons to them?***
>
> **A6.** Thank you very much for your suggestion! We agree that more discussions on the latest ANN variants need to be supplemented, and the relevant response is the same as the answer of **Q2**.
>
> ***Q7. The vector-difference operator $\tau$ plays an important role in the SVMT module. Does it capture relative geometric offsets or spike-timing discrepancies?***
>
> **A7.** Thank you for this insightful question. The vector difference operator $\tau$ ($\tau = Q - K$) is indeed central to the SVMT module, and it is designed to capture both relative geometric offsets and spike-timing discrepancies in an integrated manner, which is detailed in Appendix C of our paper. The specific mechanism is as follows:
>
> **(1) Geometric offset capture:** The query (Q) and Key (K) vectors are derived from the spiking encoding of point cloud features, which inherently represent geometric attributes like spatial coordinates. The channel-wise vector difference $\tau = Q - K$  directly quantify the relative geometric displacement between points (e.g., coordinate differences of adjacent points). This offset is then gated by spiking neurons to generate a sparse, binary attention map, allowing the module to focus precisely on salient local geometric structures.
>
> **(2) Reflection of spike-timing differences:** The temporal dynamics of SNNs integrate these geometric difference over time. The temporal dynamics of SNNs integrate these geometric differences over time. Points with similar features fire at correlated timesteps, resulting in a small, sub-threshold $\tau$. In contrast, significant geometric differences cause asynchronous firing, leading to a larger accumulated $\tau$ that exceeds the threshold and triggers a spike. Thus, while $\tau$ is computed from instantaneous values, its effect on the spike output is a direct consequence of the underlying timing differences between Q and K.
>
> In summary, $\tau$ directly encodes geometry, and the spiking mechanism translates this spatial difference into a temporally-coded signal.

---

> ### Comment · Reviewer_R7n5 · 2025-11-28
>
> Thanks for the authors' answers. My questions had been basically solved. If the paper was accepted, I hope the revised content will be updated in the camera-ready version.

---

### Comment · Area_Chair_Wm5s · 2025-11-27
**Reminder: Engage in Discussions and Finalize Your Review**

Dear Reviewers,

Thank you for your valuable reviews. With the Reviewer-Author Discussions deadline approaching, please take a moment to read the authors’ rebuttal and the other reviewers’ feedback, and participate in the discussions and respond to the authors. Finally, be sure to complete the “Final Justification” text box and update your “Rating” as needed. Your contribution is greatly appreciated. I will flag irresponsible (final) reviews and/or any reviewers not participating in discussions.

Reviewers are expected to stay engaged in discussions, initiate them, respond to authors’ rebuttal, ask questions, and listen to answers to help clarify remaining issues.

It is not OK to stay quiet.

It is not OK to leave discussions till the last moment.

If authors have resolved your (rebuttal) questions, do tell them so.

If authors have not resolved your (rebuttal) questions, do tell them so too.

Thanks.
AC

---

### Author Response · Authors · 2025-11-30
**Global Response [Update of PDF & Point-by-Point Reply]**

First and foremost, we would like to express our sincere gratitude to the Reviewers, Area Chairs, and Program Chairs for their time and dedicated effort. We have carefully studied all comments and have provided detailed point-by-point responses under each Reviewer's section.

We are encouraged not only by the positive scores but, more importantly, by the recognition of our work from the reviewers across three key dimensions:

- **Method:** "Logical and technically sound" (`Reviewer R7n5`); "not a trivial substitution of ReLU with spikes, redefines attention in spike terms" (`Reviewer NzVp`); "SVMT module is well-justified and cleverly leverages binary spiking masks" (`Reviewer j5DG`); "superior performance on classification and segmentation tasks" (`Reviewer Fv5h`); "ablation studies make a credible case for the architecture" (`Reviewer NzVp`).

- **Evaluation:** "Complete and adequate experimental results supporting claims" (`Reviewer R7n5`); "benchmarked on diverse datasets covering clean geometry, noisy scans, and part segmentation" (`Reviewer NzVp`); "significantly reduces parameter count and energy consumption" (`Reviewer Fv5h`); "solid SOTA performance among SNNs" (`Reviewer j5DG`).

- **Presentation:** "Well-organized, figures readable and understandable" (`Reviewer R7n5`); "detailed presentation of the methodology" (`Reviewer Fv5h`); "clear and logically structured" (`Reviewer R7n5`); "well-written with supplementary code" (`Reviewer Fv5h`).

In response to the reviewers’ questions and suggestions, we have made substantial updates to strengthen the manuscript, including supplementing strong ANN baselines, expanding scene-level experiments, adding quantitative validation for key modules, clarifying technical details, correcting data inconsistencies, and supplementing critical deployment metrics.

These updates, along with refined descriptions, are detailed in the revised PDF (highlighted in blue). A summary of these changes follows:

- ***Main Text Tables:***

    ○ To address Reviewer `R7n5` and `NzVp`’s concerns about strong ANN baselines, we updated Table 1 with Point-Bert and PointNeXt’s comparative data (OA, FLOPs, energy consumption) under a unified protocol.

    ○ To address Reviewer `R7n5`, `NzVp`, and `j5DG`’s requests for scene-level validation, we moved S3DIS results from the appendix to main text Table 3, and added new Table 4 (SemanticKITTI) and Table 5 (ScanNet V2) for outdoor/indoor scene segmentation.

    ○ To address Reviewer `j5DG`’s focus on deployment metrics, we added new Table 10 reporting per-sample inference latency and peak memory usage on NVIDIA A6000 GPU.

- ***Appendix Supplements & Clarifications:***

    ○ To address Reviewer `Fv5h` and `NzVp`’s questions on DSSR’s stability, we added Figure 6 (training loss curves with/without DSSR) in Appendix B, complementing existing mathematical proofs of gradient boundedness.

    ○ To address Reviewer `NzVp`’s inquiry about Spike Point Masker, we clarified the tensor layout (T×C×N), specified summation axes for point/channel masks, and added Algorithm 1-2 (pseudocode) in Appendix C.

    ○ To address Reviewer `R7n5` and `j5DG`’s questions on energy calculation, we reinforced Appendix F-H with calibrated parameters from Intel Loihi, hardware correlation explanations, and biological plausibility details.

    ○ To address Reviewer `R7n5`’s question on SVMT’s τ operator, we detailed its dual capture of geometric offsets and spike-timing discrepancies in Appendix C.

- ***Data & Format Corrections:***

    ○ To address Reviewer `Fv5h`’s concern about inconsistent energy calculations, we corrected SpikePointNet’s FLOPs (scaled to T=16) and recalculated its energy consumption for consistency in Table 1.

    ○ To address Reviewer `R7n5`’s citation format issue, we regenerated the entire bibliography using ICLR’s official BibTeX template.

- ***Terminology & Architecture Clarification:***

    ○ To address Reviewer `NzVp`’s comment on hybrid architecture ambiguity, we explicitly stated in the main text that the input embedding and FC head use MAC operations, while the encoder/decoder are spike-friendly (AC-heavy).

    ○ To address Reviewer `Fv5h`’s question on Equation 7’s subtraction rationale, we clarified it as an inverted residual connection for sparse spike features, retaining full information via subsequent integration.

We sincerely hope to engage in constructive dialogue with reviewers as we believe this exchange is vital for improving the quality of our work.

Best regards,

Authors of Submission2011

---

### Meta-Review · Area_Chair_cWFv · 2025-12-08

**Summary:**

The paper proposes a new SNN model for point cloud classification and segmentation. The reviewers raised many issues in their initial reviews, and the authors responded faithfully with a detailed rebuttal. Although many suggested experiments and clarifications have been conducted and addressed, the AC still finds that the original technical novelty of this SNN model is limited, as it largely follows early ANN architectures for similar tasks while adapting key modules to the SNN setting.

In addition, the question of “energy efficiency,” raised by multiple reviewers, is not convincingly resolved: the authors do not conduct practical measurements of energy consumption, but just estimate and theoretically predict it by referring to other literature. This makes the core efficiency claims difficult to validate.

Moreover, the authors have introduced substantial revisions (new experiments, figures, and textual changes) that would need to be carefully checked and assessed in another review cycle.

Overall, the AC recommends rejecting the paper in its current form.

**Reviewer Concerns:**

The technical novelty, the lack of direct evidence on practical energy consumption, and the large amount of substantial new revisions prevent me from recommending acceptance this time. In addition, Reviewer j5DG, who read and acknowledged the rebuttal, also expressed hesitation and did not shift to a positive recommendation.

**Reviewer Scores:**

Reviewer R7n5: 6 (confirmed and maintained)
Reviewer Fv5h: 4 (no reply)
Reviewer NzVp: 4 (no reply)
Reviewer j5DG: 4 (confirmed but hesitated to raise the score, saying "It solves my major concerns. I will raise my score if other reviewers do not have further questions.")

---

### Decision · Program_Chairs · 2026-01-26

Reject